# Directed Probabilistic Watershed

**Enrique Fita Sanmartín,      Sebastian Damrich,      Fred A. Hamprecht**
HCI/IWR at Heidelberg University, 69120 Heidelberg, Germany
`{enrique.fita.sanmartin, sebastian.damrich, fred.hamprecht}`
`@iwr.uni-heidelberg.de`

## Abstract

The Probabilistic Watershed is a semi-supervised learning algorithm applied on undirected graphs. Given a set of labeled nodes (seeds), it defines a Gibbs probability distribution over all possible spanning forests disconnecting the seeds. It calculates, for every node, the probability of sampling a forest connecting a certain seed with the considered node. We propose the "Directed Probabilistic Watershed", an extension of the Probabilistic Watershed algorithm to directed graphs. Building on the Probabilistic Watershed, we apply the Matrix Tree Theorem for directed graphs and define a Gibbs probability distribution over all incoming directed forests rooted at the seeds. Similar to the undirected case, this turns out to be equivalent to the Directed Random Walker. Furthermore, we show that in the limit case in which the Gibbs distribution has infinitely low temperature, the labeling of the Directed Probabilistic Watershed is equal to the one induced by the incoming directed forest of minimum cost. Finally, for illustration, we compare the empirical performance of the proposed method with other semi-supervised segmentation methods for directed graphs.

## 1   Introduction

In the last years, machine learning has experienced a great boost thanks to the large quantity of available data and increasing accessibility of less expensive and more powerful processing capacities. However, the acquisition of labeled data is still laborious in some applications like computer-aided diagnosis or drug discovery. Semi-supervised learning is the subfield of machine learning that utilizes both labeled and unlabeled data. It permits exploiting the large amount of unlabeled data available in many use cases jointly with frequently smaller sets of labeled data. When data is encoded as a network, graph-based semi-supervised learning aims to assign a label or class to each of the nodes of a network based on its topology and a given set of labeled nodes / seeds. Graph-based semi-supervised learning has been applied in multiple domains like computer vision [6, 15, 17], NLP [2, 28, 31], social networks [1, 45] and biomedical science[24]. Most graph-based semi-supervised learning algorithms focus on undirected weighted graphs [5, 36], though directed graphs appear naturally in many cases like in k-Nearest Neighbors graphs or citation and recommendation networks among others. Instances of methods proposed for directed graphs are [9, 11, 42, 48, 49].

Recently, [12] proposed the transductive graph-based semi-supervised learning algorithm Probabilistic Watershed (ProbWS). It envisages a Gibbs distribution over the labeled nodes separating forests (forests whose trees contain a unique labeled node) that span a weighted undirected graph. The probability of each forest is proportional to its weight. By means of the Matrix Tree Theorem [19, 40], the ProbWS computes the probability of sampling a seed separating forest such that a query node is connected to a single labeled node / seed. This approach turns out to be equivalent computationally and by result to the Random Walker / Harmonic energy minimization [3, 15, 47, 50].

The Matrix Tree Theorem [19, 40] has been generalized to directed graphs [25]. We propose an extension of the ProbWS, the "Directed Probabilistic Watershed" (DProbWS), that can be applied

35th Conference on Neural Information Processing Systems (NeurIPS 2021).

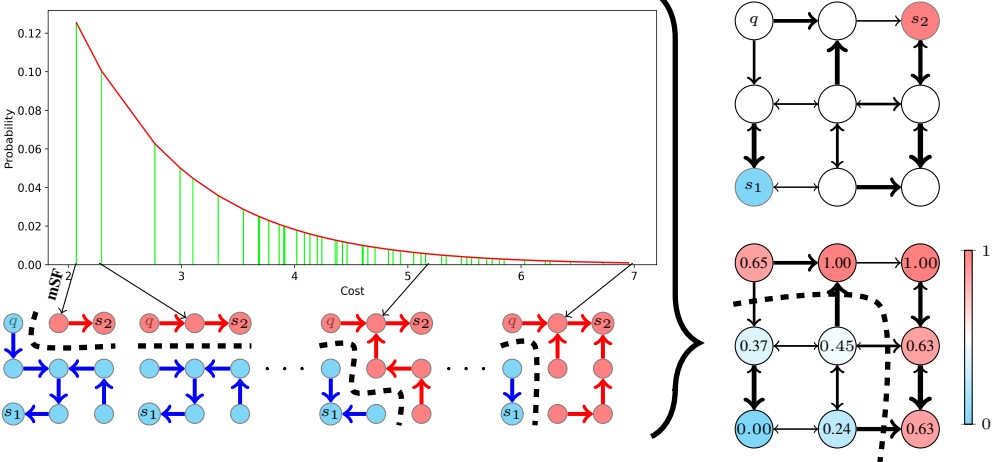

Figure 1: The DProbWS computes the expected seed assignment of every node for a Gibbs distribution over all exponentially many spanning in-forests in closed-form. It thus avoids the winner-takes-all behaviour of the minimum cost spanning in-forest (mSF). (**Top right**) Two-seeded directed graph with edge costs represented by the widths of the arrows. (**Bottom left**) The mSF and other, higher cost in-forests. The mSF assigns the query node $q$ to $s_1$. Other in-forests of low cost might however induce different labelings. (**Top left**) We therefore consider a Gibbs distribution over all spanning in-forests with respect to their cost (see equation (6)). Each green bar corresponds to the cost of one of the 60 possible spanning in-forests. (**Bottom right**) DProbWS probabilities for assigning a node to $s_2$. The dashed lines indicate the cut of the assignment. Query $q$ is now assigned to $s_2$. Considering a distribution over all spanning in-forests gives an uncertainty measure and can yield a different assignment from the mSF's. In contrast to the 60 forests in this toy graph, for real-life graphs the number of in-forests increases exponentially with the number of edges and nodes.

to directed graphs. Instead of considering seed separating spanning forests, we deal with incoming directed forests rooted at the seed nodes (see Figure 1). We analyze the DProbWS from a theoretical point of view and generalize the results presented in [12] to directed graphs. Building on the arguments used in the original ProbWS paper, we

- analytically compute the probability that a graph node is assigned to a particular seed in an ensemble of Gibbs distributed incoming directed spanning forests rooted at the seeds,

- demonstrate the equivalence of the Directed Probabilistic Watershed and the directed version of the Random Walker presented in [15],

- propose a natural extension of the Power Watershed [6] to directed graphs,

- run an illustrative experiment to show the empirical performance of the Directed Probabilistic Watershed and compare it with other semi-supervised transductive methods.

## 1.1 Related work

Semi-supervised learning is an important subfield of machine learning due to is ability to exploit both labeled and unlabeled data [41]. If data is represented as a graph, graph based semi-supervised learning exploits, in addition to the labeled samples, the topology of the graph to infer the class of the unlabeled nodes. Undirected graphs have received more attention in this field than the directed ones [5, 36]. Nonetheless, there have been methods proposed for directed graphs. In [48, 49] the labels are inferred by minimizing a regularized function that forces highly connected subgraphs to have the same label. The paper [42] builds a symmetric pairwise similarity via co-linkage analysis from the directed graph in order to apply algorithms for undirected graphs. The work presented in [11] proposes a method based on game theory, where the game's Nash equilibrium defines the labeling of the data. Related to our work by the use of absorbing random walks is the one proposed in [9]. Their algorithm assigns the label of the seed that maximizes the accumulated expected number of visits from the unlabeled query node before being absorbed by an artificially added metanode. We, instead,

prove that the DProbWS turns out to assign the label of the seed that maximizes the probability of absorption of a random walker.

Our method extends the Probabilistic Watershed (ProbWS) [12], which, in turn, is based on the Watershed algorithm [7]. The ProbWS paper established a link between the Watershed [7] and the Random Walker [3, 15, 47, 50]. Another such connection was made by the Power Watershed paper [6]. We extend these relations to the directed graph framework.

We smooth the combinatorial minimum spanning in-forest via entropic regularization. More specifically, we consider a Gibbs distribution over all spanning in-forests, as was proposed in the original ProbWS paper [12]. Entropic regularization has been applied in other situations like the randomized shortest path framework [14, 20] or optimal transport [8]. If applied in conjunction with deep networks, it allows end-to-end training [32].

The Matrix Tree Theorem (MTT) is key to the theory developed in our work. The directed version of the MTT [25] allows us to compute in closed form the weight of a set of incoming forests. The MTT has been also applied in NLP [22], biology [38] and network analysis [37, 39]. A generalization of the MTT, the Matrix Forest Theorem (MFT), was used by [4] to define a distance between the nodes of a graph. Similar to our approach, [34] defines a Gibbs distribution over the forests using the MFT.

## 2 Background

### 2.1 Notation and terminology

In this section, we introduce the main definitions and notation used in the paper. Most of them have been borrowed from [12, 25]. Let $G = (V, E, w, c)$ be a directed graph where $V$ denotes the set of nodes, $E$ the set of edges and $w$ and $c$ are functions that assign a weight $w(e) \in \mathbb{R}_{\geq 0}$ and a cost $c(e) \in \mathbb{R}$ to each edge $e \in E$. The edge $e = (i, j)$ indicates that there is a directed edge from $i$ to $j$. If $(i, j) \notin E$, we set $c\big((i, j)\big) = \infty$ and $w\big((i, j)\big) = 0$. The set $S = \{s_1, s_2\} \subset V$ will denote the set of seed / labeled nodes and $U = V \backslash S$ the unlabeled ones. In order to ease the exposition we only consider the 2-seed scenario, although the method can be generalized to the $n$-seed scenario. We will use the terms seed and labeled node interchangeably.

**Definition 1.** Let $t = (V_t, E_t)$ be a graph then we say that it is an **incoming directed spanning tree rooted at r** $\in V_t$ (in-tree for short) if: **1)** every vertex $u \neq r$ has one and only one outgoing edge; **2)** the root node $r$ does not have any outgoing edges; **3)** $t$ does not have directed cycles, i.e., there does not exist a directed path in $t$ such that the initial and final vertex are the same. Note that condition 3), in conjunction with the other conditions, implies that no cycles (directed or undirected) will be formed.

Equivalently, one can define an in-tree as a graph in which, for any vertex $u \in V_t$ there is exactly one directed path from $u$ to the root $r$, and the root does not have any out-going edges.[1] Note that an in-tree becomes a tree in the classical sense if the directions of the edges are ignored. We say that an in-tree is spanning on $G = (V, E)$ if $t$ is a subgraph of $G$ and $V_t = V$. The set of spanning in-trees of $G$ rooted at $r$ will be denoted by $\mathcal{T}^{\vec{r}}$.

Analogously, the set $\mathcal{F}_{\vec{u}}^{\vec{v}}$ represents the set of 2-in-trees spanning forests rooted at $u$ and $v$, i.e., the spanning graphs of $G$ consisting of two disjoint in-trees, such that $u$ and $v$ are the respective roots of the in-trees. Furthermore, if we consider a third node $q$, we define $\mathcal{F}_{\vec{u},q}^{\vec{v}} \subseteq \mathcal{F}_{\vec{u}}^{\vec{v}}$, as the set of all 2-in-trees spanning forests where $q$ is connected to $u$ by a directed path. In order to shorten the notation we will refer to 2-in-trees spanning forests simply as 2-in-forests or in-forests.

We define the weight of an arbitrary graph (e.g. in-forests)) as the product of the weights of all its edges, $w(G) = \prod_{e \in E} w(e)$. The weight of a set of graphs, $w(\{G_i\}_{i=0}^n)$ is the sum of the weights of the graphs $G_i$. In a similar manner, we define the cost of a graph as the sum of the costs of all its edges, $c(G) = \sum_{e \in E} c(e)$. As in the Probabilistic Watershed [12], we consider $w(e) = \exp(-\mu c(e))$, $\mu \geq 0$, which will be a consequence of the definition of a Gibbs distribution over the 2-in-forests in $\mathcal{F}_{\vec{u}}^{\vec{v}}$. Thus, a low edge-cost corresponds to a large edge-weight, and a minimum edge-cost spanning

---

[1]One can define analogously an outgoing tree (out-tree) rooted at $r$ as a tree where there is exactly one directed path from the root $r$ to any other node, and the root does not have any incoming edges.

in-forest (mSF) is equivalent to a maximum edge-weight spanning in-forest (MSF). Depending on the context, the abbreviations mSF and MSF will be also used for the undirected versions of the minimum/maximum spanning forests. Via the Directed Matrix Tree Theorem [25], we aim to compute the weight of the set of the 2-in-forests rooted at the seeds. The theorem makes use of the out-Laplacian matrix which we define next.

**Definition 2.** Given a weighted directed graph $G = (V, E, w)$ we define the **out-Laplacian** of $G$ as

$$L := D - A^\top, \tag{1}$$

where $A \in \mathbb{R}^{|V| \times |V|}$ is the vertex-adjacency matrix of $G$ represented entry-wise as $A_{ij} = w\big((i, j)\big)$ and $D$ denotes the diagonal matrix defined as $D_{ii} = \sum_{j \in V} A_{ij}$, i.e., $D_{ii}$ is the out-degree of vertex $i$.[2] For brevity, we will refer to the out-Laplacian just as Laplacian. For any $v \in V$, $L^{[v]}$ will stand for the Laplacian after removing the row and column indexed by $v$.

## 2.2 Matrix Tree Theorem

**Theorem 2.1 (MTTdir, [25]).** For any edge-weighted directed graph $G$ with an arbitrary fixed node $r \in V$ the weight of the set of incoming directed spanning trees rooted at $r$, $w(\mathcal{T}^{\overrightarrow{r}})$, is equal to[3]

$$w(\mathcal{T}^{\overrightarrow{r}}) := \sum_{t \in \mathcal{T}^{\overrightarrow{r}}} w(t) = \sum_{t \in \mathcal{T}^{\overrightarrow{r}}} \prod_{e \in E_t} w(e) = \det(L^{[r]}).$$

Note that Theorem 2.1 generalizes the original Matrix Tree Theorem [19, 40] if we consider an undirected graph as a directed graph with both directions present for each edge. In this case, any spanning tree in the undirected graph can be interpreted as an in-tree rooted at an arbitrary root $r$ once the direction of the edges has been set appropriately. Hence, for an undirected graph the choice of the root $r$ is irrelevant and we retrieve the original MTT.

We propose to follow the approach taken in [12], but with the generalization of the Matrix Tree Theorem to directed graphs and thus obtain a directed version of the ProbWS.

## 2.3 Random Walker and Power Watershed

In this section, we summarize the Random Walker [15] and Power Watershed [6], two semi-supervised graph-based algorithms which are directly connected to the Probabilistic Watershed. Both algorithms consider an undirected graph, $G$, with a set of seeds $S = \{s_1, s_2\}$, i.e., labeled nodes.

The Random Walker paper [15] addresses the following problem: *What is the probability that a random walker starting at node $q$ reaches seed $s_1$ before reaching seed $s_2$?* The solution to this question can be obtained by solving the combinatorial Dirichlet problem, which consists of finding the minimizer of

$$\frac{1}{2} x^\top L x = \frac{1}{2} \sum_{e = \{u, v\} \in E} w(e)(x_u - x_v)^2, \text{ s.t. } x_{s_1} = 1, \ x_{s_2} = 0, \tag{2}$$

where $L$ denotes the undirected Laplacian of $G$. The minimizer of equation (2) is the solution of the following linear system

$$L_U x_U^{s_i} = -B_{s_i}^\top, \tag{3}$$

where $x_U^{s_i}$ represents the probability that the unlabeled nodes in $U$ are absorbed by $s_i$, $L_U$ is the square submatrix of $L$ indexed by the elements in $U$ and $B_{s_i}^\top$ is the row $s_i$ of $L$ without the seeds.

The Power Watershed generalizes the framework of the Random Walker and minimizes the following objective function

$$\sum_{e = \{u, v\} \in E} w^\alpha(e)(x_u - x_v)^\beta, \text{ s.t. } x_{s_1} = 1, \ x_{s_2} = 0, \tag{4}$$

---

[2]Analogously, the in-Laplacian is defined as $L_{in} = D_{in} - A$, where $D_{in}$ is the diagonal matrix with the in-degrees of the nodes.

[3]The weight of the out-trees can be calculated using the in-Laplacian instead of the out-Laplacian.

for $\alpha, \beta \geq 0$. For instance, $\alpha = 1$ and $\beta = 2$ define the Random Walker's objective function. The Power Watershed analyzes the case $\alpha \to \infty$ and $\beta = 2$, and boils down to computing a mSF (minimum cost Spanning Forest) and applying the Random Walker in the plateaus (connected subgraphs with constant edge weight), resolving the ambiguity of which mSF is sampled when there is more than one.

The generalization of these algorithms to directed graphs is not straightforward if one takes the objective function approach. Solving (2) directly is equivalent to solve the problem with an undirected graph whose edge-weights are equal to the sum of the edge-weights in both directions of the original directed graph. The method proposed in [35] generalizes the combinatorial Dirichlet problem (2) by interpreting the graph as an electrical network with diodes. This approach is not equivalent to the random walk on a directed graph anymore.

The Random Walker can also be generalized to the directed case if we use the intuitive approach instead, i.e., we consider the probability of a random walker (constrained to follow the directions of the edges) being absorbed by a certain seed. In the supplemental material, we give a proof of how these probabilities can be computed in the directed case. We demonstrate that the solution of the linear system (3) with the Laplacian transposed provides the desired probabilities. In section 5, we show how the Power Watershed can be generalized to directed graphs by means of the DProbWS.

## 2.4 Probabilistic Watershed review

The Probabilistic Watershed (ProbWS) [12] is based on the Watershed algorithm. The Watershed algorithm calculates a minimum cost spanning forest, mSF, such that the seeds belong to different components [7]. A query node inherits the label of the seed to which it is connected in the mSF. The ProbWS instead, considers all possible seed-separating spanning forests. It defines a Gibbs probability distribution over the forests and analytically computes, for every node, the probability of sampling a forest connecting a certain seed with that node via the application of the Matrix Tree Theorem (for undirected graphs).

In [12] it was shown that the ProbWS turns out to be equivalent to the Random Walker algorithm proposed in [15]. Additionally, it was shown that the Power Watershed potentials [6] were given by the ProbWS probabilities when the latter is restricted to mSFs instead of all spanning forests. This restriction manifests when the entropy of the Gibbs distribution of forests is minimized.

In this work, we extend the ProbWS framework to directed graphs and call it Directed Probabilistic Wasterhsed. In this case, the spanning forests considered are formed by in-forests rooted at the seeds, i.e., directed trees with a unique path from every node to the seed. Inspired by [12], we define a Gibbs Probability distribution over the in-forests and we apply the Matrix Tree Theorem for directed graphs to compute the probability of sampling a forest connecting a certain seed with a query node in closed-form. Next, we demonstrate the equivalence to the Directed Random Walker and finally we propose an extension of the Power Watershed for directed graphs.

## 3 Directed Probabilistic Watershed

### 3.1 Gibbs probability distribution

As in [12], we define a Gibbs probability distribution over 2-in-forests with given entropy $J$, such that the 2-in-forests with a lower cost have a higher probability mass. Formally, the 2-in-forests are sampled from the distribution which minimizes

$$\min_P \sum_{f \in \mathcal{F}_{\vec{s_1}}^{\vec{s_2}}} P(f)c(f), \quad \text{s.t.} \quad \sum_{f \in \mathcal{F}_{\vec{s_1}}^{\vec{s_2}}} P(f) = 1 \text{ and } \mathcal{H}(P) = J, \tag{5}$$

where $\mathcal{H}(P)$ is the entropy of $P$. The lower the entropy, the more probability mass is given to the 2-in-forests of lowest cost. The minimizing distribution is the Gibbs distribution ([see 43, 3.2]):

$$P(f) = \frac{\exp\left(-\mu c(f)\right)}{\sum_{f' \in \mathcal{F}_{\vec{s_1}}^{\vec{s_2}}} \exp\left(-\mu c(f')\right)} = \frac{\prod_{e \in E_f} \exp(-\mu c(e))}{\sum_{f' \in \mathcal{F}_{\vec{s_1}}^{\vec{s_2}}} \prod_{e' \in E_{f'}} \exp(-\mu c(e'))} = \frac{w(f)}{\sum_{f' \in \mathcal{F}_{\vec{s_1}}^{\vec{s_2}}} w(f')}, \tag{6}$$

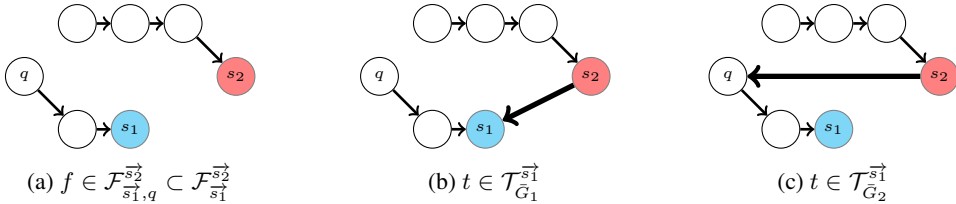

(a) $f \in \mathcal{F}_{\vec{s1},q}^{\vec{s2}} \subset \mathcal{F}_{\vec{s1}}^{\vec{s2}}$      (b) $t \in \mathcal{T}_{\bar{G}_1}^{\vec{s1}}$      (c) $t \in \mathcal{T}_{\bar{G}_2}^{\vec{s1}}$

Figure 2: Procedure to transform an in-forest in $\mathcal{F}_{\vec{s1}}^{\vec{s2}}$ to a spanning in-tree in $\bar{G}_1$ and $\bar{G}_2$ as defined in Lemma 3.3. (2a) In-forest $f \in \mathcal{F}_{\vec{s1},q}^{\vec{s2}} \subseteq \mathcal{F}_{\vec{s1}}^{\vec{s2}}$. (2b) Transformation of $f$ into an in-tree $t \in \mathcal{T}_{\bar{G}_1}^{\vec{s1}}$ after adding the edge $(s_2, s_1)$ (thick edge). (2c) Transformation of $f$ into an in-tree $t \in \mathcal{T}_{\bar{G}_2}^{\vec{s1}}$ after adding the edge $(s_2, q)$ (thick edge).

where $\mu$ can be interpreted as the inverse temperature of the Gibbs distribution. The parameter $\mu$ implicitly determines the entropy. A higher $\mu$ (lower temperature) implies a lower entropy. Equation (6) motivates our choice of edge weights $w(e) = \exp(-\mu c(e))$.

**Definition 3** (**Probabilities of the Directed Probabilistic Watershed**). Given two seeds $s_1$ and $s_2$ and a query node $q$, we define the Directed Probabilistic Watershed's probability that $q$ and $s_1$ have the same label as the probability of sampling a 2-in-forest from $\mathcal{F}_{\vec{s1}}^{\vec{s2}}$ that connects $q$ to $s_1$ by a directed path, i.e., they belong to the same in-tree rooted at $s_1$:

$$P(q \sim s_1) := \sum_{f \in \mathcal{F}_{\vec{s1},q}^{\vec{s2}}} P(f) = \sum_{f \in \mathcal{F}_{\vec{s1},q}^{\vec{s2}}} w(f) \Big/ \sum_{f' \in \mathcal{F}_{\vec{s1}}^{\vec{s2}}} w(f') = w\left(\mathcal{F}_{\vec{s1},q}^{\vec{s2}}\right) \Big/ w\left(\mathcal{F}_{\vec{s1}}^{\vec{s2}}\right). \quad (7)$$

The Directed Probabilistic Watershed (DProbWS) takes all spanning 2-in-forests into account according to their cost (see Figure1). The resulting assignment probability of each node provides an uncertainty measure. Assigning each node to the seed for which it has the highest probability can yield a labeling different from the in-forest with highest individual probability, the mSF.

**Remark 3.1.** Note that the probability given by the DProbWS is well defined if and only if $w\left(\mathcal{F}_{\vec{s1}}^{\vec{s2}}\right) \neq 0$, that is if and only if there exists at least one in-forest rooted at the seeds. As far as any node is connected by a directed path to at least one of the seeds, such an in-forest will exist. If the seeds were not reachable from a node $q$, this node would have no connection with the seeds and therefore we could not infer any label. These nodes are termed zero-knowledge nodes. We can still use the proposed method by removing all nodes in the zero-knowledge components [30]. In the remaining we will assume $w\left(\mathcal{F}_{\vec{s1}}^{\vec{s2}}\right) \neq 0$.

**Remark 3.2.** Note that the DProbWS probabilities have been based on the concept of incoming forests. The same reasoning can be extended for outgoing forests just by reversing the edge directions.

### 3.2 Computation of the Directed Probabilistic Watershed probabilities

In this section, we explain how to compute the probabilities defined by the DProbWS. While for undirected graphs it is possible to compute $w(\mathcal{F}_{s_1}^{s_2})$ as a difference of the weight of the set of spanning trees of the graph $G$ and the weight of the set of spanning trees of G after adding an edge connecting the seeds, this is not possible for $w(\mathcal{F}_{s_1,q}^{s_2})$.[4] In [12], $w(\mathcal{F}_{s_1,q}^{s_2})$ is obtained by solving a linear system involving $w(\mathcal{F}_{s_1}^{s_2})$, $w(\mathcal{F}_{s_2}^{q})$ and $w(\mathcal{F}_{s_1}^{q})$. In contrast to the undirected case, we can bypass the auxiliary linear system due to the directions of the edges which restrict the number of possible directed forests. This permits us to express $w(\mathcal{F}_{\vec{s1}}^{\vec{s2}})$ and $w(\mathcal{F}_{\vec{s1},q}^{\vec{s2}})$ in terms of the weight of the set of in-trees of an augmented graph as it is shown in the following lemma.

**Lemma 3.3.** Let $G = (V, E, w)$ be a directed graph. Given $s_1, s_2, q \in V$, define the graphs $\bar{G}_1 = (V, E \cup (s_2, s_1), \bar{w}_1)$ and $\bar{G}_2 = (V, E \cup (s_2, q), \bar{w}_2)$ where $\bar{w}_1(e) = \bar{w}_2(e) = w(e) \; \forall e \in E$ and $\bar{w}_1((s_2, s_1)) = \bar{w}_2((s_2, q)) = 1$. Then

$$\text{a) } w\left(\mathcal{F}_{\vec{s1}}^{\vec{s2}}\right) = w\left(\mathcal{T}_{\bar{G}_1}^{\vec{s1}}\right) - w\left(\mathcal{T}_{G}^{\vec{s1}}\right) \qquad \text{b) } w\left(\mathcal{F}_{\vec{s1},q}^{\vec{s2}}\right) = w\left(\mathcal{T}_{\bar{G}_2}^{\vec{s1}}\right) - w\left(\mathcal{T}_{G}^{\vec{s1}}\right).$$

---

[4] $\mathcal{F}_{s_1,q}^{s_2}$, $\mathcal{F}_{s_1}^{s_2}$, $\mathcal{F}_{s_2}^{q}$ and $\mathcal{F}_{s_1}^{q}$ are equivalents to $\mathcal{F}_{\vec{s1},q}^{\vec{s2}}$, $\mathcal{F}_{\vec{s1}}^{\vec{s2}}$, $\mathcal{F}_{\vec{s2}}^{\vec{q}}$ and $\mathcal{F}_{\vec{s1}}^{\vec{q}}$ but defined in an undirected graph.

**Proof:** We will only prove **a)**. Note that $w\left(\mathcal{T}_{\bar{G}_1}^{\overrightarrow{s_1}}\right) - w\left(\mathcal{T}_{G}^{\overrightarrow{s_1}}\right)$ is the weight of all the spanning trees of $\bar{G}_1$ rooted at $s_1$ containing the edge $(s_2, s_1)$. It is easy to see that for any $f \in \mathcal{F}_{\overrightarrow{s_1}}^{\overrightarrow{s_2}}$, there exists a unique $t \in \mathcal{T}_{\bar{G}_1}^{\overrightarrow{s_1}}$ containing $(s_2, s_1)$ such that $w(t) = w(f)$. Indeed, if we add the edge $(s_2, s_1)$ to $f$ we obtain a tree $t \in \mathcal{T}_{\bar{G}_1}^{\overrightarrow{s_1}}$ since for every node in $V$, there exists a unique path to $s_1$ (see Figure 2b). Conversely, any tree in $t \in \mathcal{T}_{\bar{G}_1}^{\overrightarrow{s_1}}$ containing the edge $(s_2, s_1)$ will be transformed into a forest $f \in \mathcal{F}_{\overrightarrow{s_1}}^{\overrightarrow{s_2}}$ after the removal of the edge of $(s_2, s_1)$. Due to $\bar{w}_1\big((s_2, s_1)\big) = 1$ the equality of weights $w(f) = w(t)$ follows. The argument for **b)** is analogous to the previous case (see Figure 2c).   $\square$

The following result (Lemma 3.4) provides a closed formula for $w\left(\mathcal{F}_{\overrightarrow{s_1}}^{\overrightarrow{s_2}}\right)$ and $w\left(\mathcal{F}_{\overrightarrow{s_1},q}^{\overrightarrow{s_2}}\right)$ in terms of the in-trees of $G$ rooted at the seeds. The proof makes use of the MTTdir (Theorem 2.1), Lemma 3.3 and the well-known Determinant Lemma [16] (Theorem A.1 appendix). Due to its technicality, we defer the proof to the supplemental material.

**Lemma 3.4.** Let $G = (V, E, w)$ be a directed graph. Let $l_{ij}^{-1,[v]}$ represent the entry $ij$ of the matrix $\left(L^{[v]}\right)^{-1}$ for any $v \in V$. Given $s_1$, $s_2$ and $q$:

**a)** $w\left(\mathcal{F}_{\overrightarrow{s_1}}^{\overrightarrow{s_2}}\right) = w(\mathcal{T}^{\overrightarrow{s_1}})l_{s_2 s_2}^{-1,[s_1]} = w(\mathcal{T}^{\overrightarrow{s_2}})l_{s_1 s_1}^{-1,[s_2]},$   **b)** $w\left(\mathcal{F}_{\overrightarrow{s_1},q}^{\overrightarrow{s_2}}\right) = w(\mathcal{T}^{\overrightarrow{s_1}})(l_{s_2 s_2}^{-1,[s_1]} - l_{s_2 q}^{-1,[s_1]}).$

As a consequence of Lemma 3.4, we can conveniently find a closed form for the probabilities of the DProbWS (Definition 3):

**Theorem 3.5.** The Directed Probabilistic Watershed probabilities are equal to

$$\Pr(q \sim s_1) = \frac{l_{s_2 s_2}^{-1,[s_1]} - l_{s_2 q}^{-1,[s_1]}}{l_{s_2 s_2}^{-1,[s_1]}} \text{ and } \Pr(q \sim s_2) = \frac{l_{s_2 q}^{-1,[s_1]}}{l_{s_2 s_2}^{-1,[s_1]}},$$

or equivalently

$$\Pr(q \sim s_2) = \frac{l_{s_1 s_1}^{-1,[s_2]} - l_{s_1 q}^{-1,[s_2]}}{l_{s_1 s_1}^{-1,[s_2]}} \text{ and } \Pr(q \sim s_1) = \frac{l_{s_1 q}^{-1,[s_2]}}{l_{s_1 s_1}^{-1,[s_2]}}.$$

Our discussion was constrained to the case of two seeds only to ease our explanation. We can reduce the case of multiple seeds per label to the two seed case by merging all nodes seeded with the same label. Similarly, the case of more than two labels can be reduced to the two label scenario by using a "one versus all strategy": We choose one label and merge the seeds of other labels into one unique seed. In both cases we might add multiple edges between node pairs. While having formulated our arguments for simple graphs, they are also valid for multigraphs by the same arguments as in [12].

## 4   Equivalence of DProbWS and the Directed Random Walker

Resembling the ProbWS equivalence with the Random Walker, the DProbWS also turns out to be equivalent to the directed version of the Random Walker. Although multiple references exist on how to compute the absorbing probabilities of a random walker in terms of the Laplacian matrix for undirected graphs [13, 15], we could not find any suitable reference for the directed case.[5] We sketch how the absorbing probabilities can be obtained for the directed case in terms of the Laplacian matrix and refer to the supplemental material for a complete derivation.

**Theorem 4.1.** The probability, $x_q^{s_1}$, that a random walker on a directed graph starting at node $q$ first reaches $s_1$ before reaching $s_2$ is given by the solution of the following linear system

$$L_U^\top x_U^{s_1} = -\left[B_1^\top\right]_{s_1}, \tag{8}$$

where $x_U^{s_i}$ represents the probability that the nodes in $U$ are absorbed by $s_i$, $L_U$ is the square submatrix of $L$ indexed by the elements in $U$ and $\left[B_1^\top\right]_{s_1}$ is the column $s_1$ of $L$ without the entries indexed by the seeds.

---

[5]However, there exist closed formulas to compute the absorbing probabilities of Markov chains in terms of the fundamental matrix [18].

**Proof sketch:** Let $P$ denote the transition probability matrix defined by the absorbing random walker. We note that the probability of being absorbed by $s_1$ at $q$ can be obtained by computing the entry $(q, s)$ of $P^n$ when $n$ tends to infinity, i.e., $x_q^{s_1} = \lim_{n\to\infty} [P^n]_{qs}$. By expressing the transition matrix in terms of the Laplacian, we find a closed formula of the matrix $P^n$ by induction. By taking the limit to infinity, we obtain the desired result. See the supplemental material for further details. $\square$

**Remark 4.2.** Note that (8) is the transposed version of the linear system solved by the undirected Random Walker and equivalently the ProbWS (3), i.e., $(L_U) y_U = -B_1^\top$. Since [12, 15] consider an undirected graph $(L_U^\top) = L_U$, the transposition becomes irrelevant in the undirected case.

The next theorem states the equivalence between the DProbWS and the Directed Random Walker. The proof is analogous to Theorem 4.1 in [12] and we replicate the proof in the supplemental material.

**Theorem 4.3.** The probability, $x_q^{s_1}$, that a random walker on a directed graph starting at node $q$ first reaches $s_1$ before reaching $s_2$ is equal to the Directed Probabilistic Watershed's probability (Definition 3)

$$x_q^{s_1} = P(q \sim s_1).$$

The equivalence stated by Theorem 4.3 combined with Theorem 4.1 implies that in order to obtain the DProbWS probabilities defined in Theorem 3.5, it is just necessary to solve a linear system for each seed node (see Algorithm 1), as it is the case for the Random Walker algorithm.

---

**Algorithm 1:** DProbWS

---

**Input:** $G = (V, E, w)$, seeds
**Output:** $x$                                                      // DProbWS probabilities
  **1** $U = V \backslash$ seeds                                              // Unlabeled nodes
  **2** $L$=outLaplacian($G$)
  **3** $L_U = L[U; U]$   // squared submatrix of $L$ indexed by the unlabeled nodes
  **4 for** $s \in$ seeds **do**
  **5**     $B_s = L[U; s]$            // $s$ column of $L$ restricted to unlabeled nodes
  **6**     $x[U; s]$=solve($L_U^\top, -B_s$)        // Solves the linear system $L_U^\top * x_U^s = -B_s$

---

## 5 Directed Power Watershed

In the ProbWS paper [12], it was proven that the Power Watershed [6] is equivalent to applying the ProbWS restricted to the minimum cost spanning forests. This restriction corresponds to the case of a Gibbs distribution of minimal entropy over the forests. In this section, we will prove the analogous result for the DProbWS: When the entropy of the Gibbs distribution over the directed in-forests (3.1) is minimal, then DProbWS is restricted to the minimum cost spanning in-forests (mSF). This permits us to define a natural extension of the Power Watershed to directed graphs.

In Section 3.1, we defined the weight of an edge $e$ as $w(e) = \exp(-\mu c(e))$, where $c(e)$ was the edge-cost and $\mu$ implicitly determined the entropy of the 2-in-forest distribution. When $\mu \to \infty$, the distribution will have minimal entropy. Consequently, only for minimum cost spanning in-forests, mSFs (or analogously the maximum weight spanning in-forests, MSFs) the limit does not yield a zero probability, so that only they will be considered in the sampling. The proof can be found in the supplemental material.

**Theorem 5.1.** Given two seeds $s_1$ and $s_2$. Let further $w_{\max} := \max_{f \in \mathcal{F}_{\vec{s_1}}^{\vec{s_2}}} w(f)$. If the entropy of the Gibbs distribution over the in-forests is minimized (6), then

$$x_q^{s_1} = \frac{\left| \{f \in \mathcal{F}_{\vec{s_1}, q}^{\vec{s_2}} \; : \; w(f) = w_{\max}\} \right|}{\left| \{f \in \mathcal{F}_{\vec{s_1}}^{\vec{s_2}} \; : \; w(f) = w_{\max}\} \right|}.$$

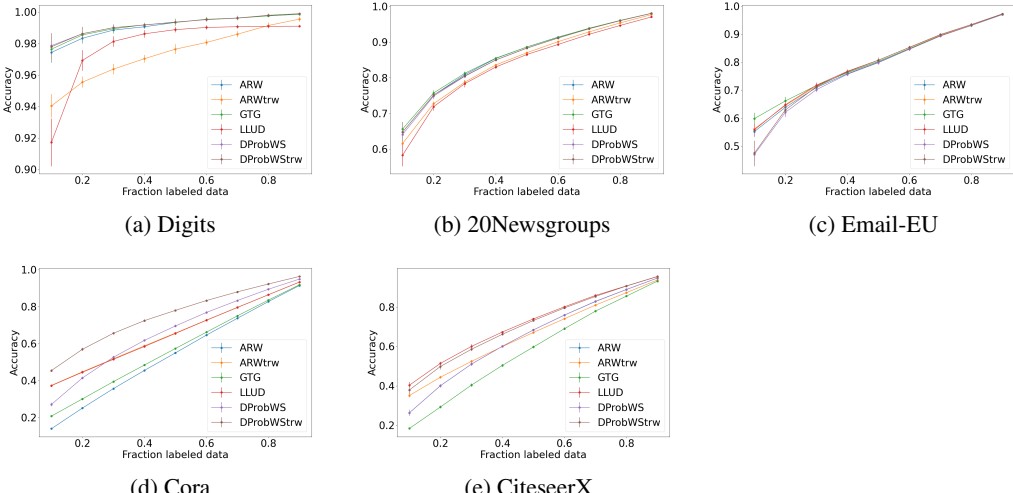

(a) Digits     (b) 20Newsgroups     (c) Email-EU

(d) Cora       (e) CiteseerX

Figure 3: Accuracy comparisons on the *Digits*, *20Newsgroups Email-EU*, *Cora* and *CiteseerX* datasets. In all plots, the horizontal axis denotes the fraction of labeled notes in the graph varying from 0.1 to 0.9 with 0.1 increment. We depict the average accuracy over 20 runs with error bars of one standard deviation. Our methods DProbWS and DProbWStrw are on par with existing methods and sometimes outperfrom them.

## 6 Experiments

Although our work has been motivated from a theoretical perspective, we run an illustrative experiment[6] to show the performance of the DProbWS on node classification. Given a graph with some labeled nodes, we infer the label of the rest of nodes. k-Nearest Neighbors (kNN) graphs are often used in practice. Due to its asymmetric nature, they can be regarded as directed graphs. We construct kNN graphs, with $k = 5$, from the UCI datasets [10] *Digits* [44] and *20Newsgroups*[23]. Additionally we consider the *Email-EU*network [26, 27, 46], the *Cora* network[29] and *CiteseerX* network[33]. We compare the DProbWS with the methods exposed in [9, 11, 49] referred as ARW, GTG and LLUD respectively. More details about the datasets, the comparison methods and the construction of the graphs can be found in the supplemental material. Due to the high sparsity of the Cora and CiteseerX networks, many of the nodes are zero-knowledge nodes and therefore its label can not be inferred (see Remark 3.1). To remedy this, we make use of the teleporting random walker (TRW) as a replacement of the natural one, in the same way it is applied in [49] (see supplemental material). The methods ARWtrw and DprobWStrw will refer to the versions that make use of the TRW. To evaluate the methods we use the accuracy (number of correctly labeled nodes divided by the total number of nodes). In the computation of the accuracy, we also include the zero-knowledge nodes. Inspired by [9], we sample a certain fraction $r$ of all nodes from each class uniformly as seeds. In Figure 3, we show the average accuracy over 20 runs for each of the $r$ values between $0.1$ and $0.9$. We observe that our method obtains comparable empirical results sometimes outperforming all competing methods.

## 7 Conclusion

In this work, we presented an extension of the Probabilistic Watershed algorithm [12] that can be applied to directed graphs.

Following the rationale exposed in the Probabilistic Watershed article [12], we defined a Gibbs distribution over all the directed spanning in-forests rooted at the seeds of a directed graph (Definition 3). We also demonstrated, using the directed version of the Matrix Tree Theorem [25], that the Directed Probabilistic Watershed is computationally and also by result equivalent to the Directed Random Walker as in the case for the undirected version. Finally, we showed that when the entropy of the

---

[6]Code publicly available at `https://github.com/hci-unihd/Directed_Probabilistic_Watershed.git`

Gibbs distribution is minimized, we obtain an extension of the Power Watershed potentials [6] to directed graphs.

In the original Probabilistic Watershed paper [12] it was showed that the effective resistance distance [21] triangle inequality gap of the triangle given by the two seeds and the query node is proportional to the probabilities of the Random Walker / ProbWS. In contrast, we have not been able to prove that the probabilities defined by the DProbWS are proportional to the triangle inequality gap of the effective resistance or any other distance in the directed setting. However, we conjecture that there may exist a distance / dissimilarity on the directed graph such that the DProbWS probability is proportional to the triangle inequality defined by the seeds and the query node. This conjecture is left for future work.

## 8    Broader impact

Our work has been motivated by a theoretical understanding of already existing algorithms. The speculation of a broader impact on this kind of works is therefore less applicable. Nonetheless, we believe that the extension of the Probabilistic Watershed to directed graphs may enrich the graph-based semi-supervised learning field, where less effort has been invested in the directed framework. Furthermore, although we believe that the Directed Random Walker extension presented here must be known, we have not found any reference applying it. By delving into the relations between the spanning in-forests and the Directed Random Walker, we expect that a better understanding of the algorithm is gained and also show the community that such extension for the directed setting exists.

From a practical point of view, the application of the Directed Probabilistic Watershed / Directed Random Walker used in conjunction with other machine learning technology may be used for many applications, both good and bad. In general, any progress on semi-supervised learning will have these same consequences.

### Acknowledgments and Disclosure of Funding

This work is supported by the Deutsche Forschungsgemeinschaft (DFG, German Research Foundation) under Germany's Excellence Strategy EXC 2181/1 - 390900948 (the Heidelberg STRUCTURES Excellence Cluster).

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
