# Directed Probabilistic Watershed
# Supplemental Material

**Enrique Fita Sanmartín,      Sebastian Damrich,      Fred A. Hamprecht**
HCI/IWR at Heidelberg University, 69120 Heidelberg, Germany
{enrique.fita.sanmartin, sebastian.damrich, fred.hamprecht}
@iwr.uni-heidelberg.de

# A    Computation of $w\left(\mathcal{F}_{\overrightarrow{s_1}}^{\overrightarrow{s_2}}\right)$ and $w\left(\mathcal{F}_{\overrightarrow{s_1},q}^{\overrightarrow{s_2}}\right)$

In this section, we prove Lemma 3.4 of the main paper, which we rename here as Lemma $A.2$. It provides a closed form for $w\left(\mathcal{F}_{\overrightarrow{s_1}}^{\overrightarrow{s_2}}\right)$ and $w\left(\mathcal{F}_{\overrightarrow{s_1},q}^{\overrightarrow{s_2}}\right)$ in terms of the in-trees of $G$ rooted at the seeds. We will use the Matrix Tree Theorem extension for directed graphs (Theorem 2.1 in the main paper), referred to as MTTdir, and the Determinant Lemma (Lemma $A.1$) which we state without proof.

**Lemma A.1** (Determinant Lemma [7]).   Given an invertible matrix $A \in \mathbb{R}^{m \times m}$ and $u, v \in \mathbb{R}^m$ we have
$$\det(A + uv^\top) = \det(A)(1 + v^\top A^{-1} u).$$

**Lemma A.2.** Let $G = (V, E, w)$ be a directed graph. Let $l_{ij}^{-1,[v]}$ represent the entry $ij$ of the matrix $\left(L^{[v]}\right)^{-1}$ for any $v \in V$. Given $s_1$, $s_2$ and $q$:

1. The weight of all 2-in-forests rooted at $s_1$ and $s_2$ is
$$w\left(\mathcal{F}_{\overrightarrow{s_1}}^{\overrightarrow{s_2}}\right) = w(\mathcal{T}^{\overrightarrow{s_1}})l_{s_2 s_2}^{-1,[s_1]} = w(\mathcal{T}^{\overrightarrow{s_2}})l_{s_1 s_1}^{-1,[s_2]}.$$

2. The weight of all 2-in-forests rooted at $s_1$ and $s_2$ such that $q$ is connected to $s_1$ is
$$w\left(\mathcal{F}_{\overrightarrow{s_1},q}^{\overrightarrow{s_2}}\right) = w(\mathcal{T}^{\overrightarrow{s_1}})(l_{s_2 s_2}^{-1,[s_1]} - l_{s_2 q}^{-1,[s_1]}).$$

**Proof:** Let $\mathbf{1}_j$ denote the column $j$ of the identity matrix $I$. Given a column vector $v \in \mathbb{R}^{|V|}$, $v^{[s_1]}$ denotes the vector $v$ after removing the entry indexed by $s_1$.

1. Let $\bar{G}_1 = (V, E \cup (s_2, s_1), \bar{w})$ be defined as in Lemma 3.3. We can compute
$$
\begin{aligned}
w\left(\mathcal{F}_{\overrightarrow{s_1}}^{\overrightarrow{s_2}}\right) &\underbrace{=}_{\text{Lemma 3.3}} w\left(\mathcal{T}_{\bar{G}_1}^{\overrightarrow{s_1}}\right) - w\left(\mathcal{T}_G^{\overrightarrow{s_1}}\right) \underbrace{=}_{\text{MTTdir}} \det\left(L_{\bar{G}_1}^{[s_1]}\right) - \det\left(L_G^{[s_1]}\right) \\
&= \det\left(L_G^{[s_1]} + \mathbf{1}_{s_2}^{[s_1]}\left(\mathbf{1}_{s_2}^{[s_1]}\right)^\top\right) - \det\left(L_G^{[s_1]}\right) \\
&\underbrace{=}_{\text{Lemma } A.1} \det(L_G^{[s_1]})\left(1 + \left(\mathbf{1}_{s_2}^{[s_1]}\right)^\top \left(L_G^{[s_1]}\right)^{-1} \mathbf{1}_{s_2}^{[s_1]}\right) - \det(L_G^{[s_1]}) \\
&= \det(L_G^{[s_1]})\ell_{s_2 s_2}^{-1,[s_1]} \underbrace{=}_{\text{MTTdir}} w(\mathcal{T}_G^{\overrightarrow{s_1}})\ell_{s_2 s_2}^{-1,[s_1]}.
\end{aligned}
\tag{1}
$$

To prove $w\left(\mathcal{F}_{\overrightarrow{s_1}}^{\overrightarrow{s_2}}\right) = w(\mathcal{T}^{\overrightarrow{s_2}})l_{s_1 s_1}^{-1,[s_2]}$ we just need to exchange the role of $s_1$ and $s_2$ in Lemma 3.3 of the main paper and follow the same steps as above.

35th Conference on Neural Information Processing Systems (NeurIPS 2021), Sydney, Australia.

2. Let $b = \mathbf{1}_{s_2} - \mathbf{1}_q$ and $\bar{G}_2$ be defined as in Lemma 3.3 of the main paper. Then

$$
w\left(\mathcal{F}_{\vec{s_1},q}^{\vec{s_2}}\right) \underbrace{=}_{\text{Lemma 3.3}} w\left(\mathcal{T}_{\bar{G}_2}^{\vec{s_1}}\right) - w\left(\mathcal{T}_G^{\vec{s_1}}\right) \underbrace{=}_{\text{MTTdir}} \det\left(L_{\bar{G}_2}^{[s_1]}\right) - \det\left(L_G^{[s_1]}\right)
$$

$$
= \det\left(L_G^{[s_1]} + b^{[s_1]}\left(\mathbf{1}_{s_2}^{[s_1]}\right)^\top\right) - \det\left(L_G^{[s_1]}\right)
$$

$$
\underbrace{=}_{\text{Lemma } A.1} \det(L_G^{[s_1]})\left(1 + \left(\mathbf{1}_{s_2}^{[s_1]}\right)^\top\left(L_G^{[s_1]}\right)^{-1} b^{[s_1]}\right) - \det(L_G^{[s_1]}) \tag{2}
$$

$$
= \det(L_G^{[s_1]})(l_{s_2 s_2}^{-1,[s_1]} - l_{s_2 q}^{-1,[s_1]}) \underbrace{=}_{\text{MTTdir}} w(\mathcal{T}^{\vec{s_1}})(l_{s_2 s_2}^{-1,[s_1]} - l_{s_2 q}^{-1,[s_1]}).
$$

$\square$

# B   Directed Random Walker

The Random Walker by Grady [6] answers the following question for undirected graphs: *What is the probability that a random walker starting at node $q$ reaches seed $s_1$ before reaching $s_2$?* If we considered the seeds as a set of absorbing nodes, then the probabilities that the Random Walker computes are the absorbing probabilities of the seeds. The Random Walker can easily be extended to the directed setting if we compute these probabilities for a directed graph. In this section, we derive the absorption probabilities of a set of seeds for a directed graph. We expect this result to be well known but we reproduce it here for the convenience of the reader since we could not find any suitable reference that expressed these probabilities in terms of the Laplacian matrix. $G = (V, E, w)$ will stand for a directed graph and $S \subset V$ will be the set of seed/labeled nodes and $U = V \backslash S$ the unlabeled nodes.

Let $L$ denote the Laplacian matrix of $G$ as defined in Definition 2 of the main paper. We index the Laplacian matrix block-wise in terms of the unlabeled and labeled
nodes in the following form

$$
L = \left( \begin{array}{cc} L_S & B_1 \\ B_2 & L_U \end{array} \right). \tag{3}
$$

**Remark B.1. Invertibility of $\mathbf{L_U}$:** Assume that for any $u \in U$, there exists some $s \in S = V \backslash U$ such that there is a directed path from $u$ to $s$. Let $\hat{G}$ denote the graph formed after merging all the vertices of $S$ into one node $s^*$. By assumption, any node $u \in U$ can reach at least one seed in $S$, therefore $s^*$ will be reachable from any node. Consequently, there exists at least one incoming tree in $\hat{G}$ rooted at $s^*$. Due to the Directed Matrix Tree Theorem, $\det(L_{\hat{G}}^{[s^*]}) = \det(L_U) \neq 0$, which implies that $L_U$ is non-singular.

In the light of the previous remark we assume that any node $u \in U$ can reach at least one of the seeds $s \in S$ via a directed path. Otherwise, if a node could not reach any seed, then it will not have a well defined directed random walker probability.

For the moment, let us assume that none of the seeds is absorbing, i.e., every seed has at least one out-going edge. Let

$$
P := D^{-1}A = I - D^{-1}L^\top \tag{4}
$$

be the transition probability matrix where $D$ and $A$ are defined as in the main text. Note that $D$ is invertible since we assume that the seeds are not absorbing and any node can reach one of the seeds, which implies that the out-degree of any node is greater than zero. The entry $P_{ij}$ denotes the probability of transitioning from node $i$ to node $j$ in one hop. Note that $\sum_{j \in V} P_{ij} = 1$ for all $i$.

We can express, in conjunction with equation (4), the transpose of the transition probability matrix as

$$
P^\top = \left( \begin{array}{cc} I_S - L_S D_S^{-1} & -B_1 D_U^{-1} \\ -B_2 D_S^{-1} & I_U - L_U D_U^{-1} \end{array} \right).
$$

Now consider the set of seeds $S$ as absorbing nodes, i.e., once a random walker reaches one of the seeds the random walker will vanish. Hence, the transition probability matrix will have the following form

$$\bar{P}^\top = \begin{pmatrix} I_S & -B_1 D_U^{-1} \\ 0 & I_U - L_U D_U^{-1} \end{pmatrix}$$

Theorem $B.3$ provides a closed formula for $\lim_{n\to\infty} (\bar{P}^n)^\top$ which will give us the absorbing probabilities of the seeds. In order to prove it, we need to state a series of definitions and results that we state without proof [see 8, Chapter 5.6].

**Definition 1.** Given an arbitrary matrix $A \in \mathbb{R}^{m\times m}$, we define the 1-norm matrix of $A$ as

$$||A||_1 = \max_j \sum_i^m |A_{ij}|.$$

**Lemma B.2.** [see 8, Corollary 5.6.16] Given an arbitrary matrix $A \in \mathbb{R}^{m\times m}$, if $||A||_1 < 1$ then $(A - I)$ is invertible and

$$(I - A)^{-1} = \sum_{i=0}^\infty A^i. \tag{5}$$

Now we can prove the following result.

**Theorem B.3.** Let $\bar{P}$ be defined as before then

$$\lim_{n\to\infty} (\bar{P}^\top)^n = \begin{pmatrix} I_S & -B_1 L_U^{-1} \\ 0 & 0 \end{pmatrix}.$$

**Proof:** Let us first prove by induction that

$$(\bar{P}^\top)^n = \begin{pmatrix} I_S & -B_1 D_U^{-1} \sum_{i=0}^{n-1} \left(I_U - L_U D_U^{-1}\right)^i \\ 0 & \left(I_U - L_U D_U^{-1}\right)^n \end{pmatrix}.$$

It naturally holds for $n = 1$. By induction

$$(\bar{P}^\top)^{n+1} = (\bar{P}^\top)^n \bar{P}^\top = \begin{pmatrix} I_S & -B_1 D_U^{-1} \sum_{i=0}^{n-1} \left(I_U - L_U D_U^{-1}\right)^i \\ 0 & \left(I_U - L_U D_U^{-1}\right)^n \end{pmatrix} \begin{pmatrix} I_S & -B_1 D_U^{-1} \\ 0 & I_U - L_U D_U^{-1} \end{pmatrix}$$

$$= \begin{pmatrix} I_S & -B_1 D_U^{-1} - B_1 D_U^{-1} \sum_{i=0}^{n-1} \left(I_U - L_U D_U^{-1}\right)^{i+1} \\ 0 & \left(I_U - L_U D_U^{-1}\right)^{n+1} \end{pmatrix}$$

$$= \begin{pmatrix} I_S & -B_1 D_U^{-1} \sum_{i=0}^{n} \left(I_U - L_U D_U^{-1}\right)^i \\ 0 & \left(I_U - L_U D_U^{-1}\right)^{n+1} \end{pmatrix}$$

Now we will prove the case when $n \to \infty$. Let as assume that the limit exist. We express it blockwise as

$$\lim_{n\to\infty} (\bar{P}^\top)^n = \begin{pmatrix} I_S & \bar{P}_1^\top \\ 0 & \bar{P}_2^\top \end{pmatrix}.$$

Since every node $u \in U$ can reach a seed node, $s$, in a finite number of hops, there exists a number of steps, $k' > 0$, such that for all $k \geq k'$, the probability, $P_{su}(k)$, of being in seed $s$ at step $k$, is greater than 0. Hence, $M_k := \left(I_U - L_U D_U^{-1}\right)^k$ is a substochastic matrix and we have

$$\left|\left|\left(I_U - L_U D_U^{-1}\right)^k\right|\right|_1 = \max_j \sum_i \left[\left(I_U - L_U D_U^{-1}\right)^k\right]_{ij} \underbrace{=}_{\substack{j^* \\ \text{maximizer}}} 1 - \underbrace{\sum_{s\in S} P_{sj^*}(k)}_{>0} < 1.$$

Therfore,

$$\lim_{n\to\infty}||M_k^n||_1 = \lim_{n\to\infty}\left|\left|\left(I_U - L_U D_U^{-1}\right)^{n\cdot k}\right|\right|_1 = \lim_{n\to\infty}\left|\left|\left(I_U - L_U D_U^{-1}\right)^{n}\right|\right|_1$$

$$\leq \lim_{n\to\infty}\left(||I_U - L_U D_U^{-1}||_1\right)^n = 0 \Rightarrow \bar{P}_2^\top = \lim_{n\to\infty}\left(I_U - L_U D_U^{-1}\right)^n = 0.$$

Furtheremore, as a consequence of Lemma $B.2$

$$\sum_{i=0}^{\infty}\left(I_U - L_U D_U^{-1}\right)^i = \left(I_U - I_U + L_U D_U^{-1}\right)^{-1} = \left(L_U D_U^{-1}\right)^{-1} = D_U L_U^{-1}.$$

Finally,

$$\bar{P}_1^\top = \lim_{n\to\infty} -B_1 D_U^{-1}\sum_{i=0}^{n}\left(I_U - L_U D_U^{-1}\right)^i = -B_1 D_U^{-1}\sum_{i=0}^{\infty}\left(I_U - L_U D_U^{-1}\right)^i$$

$$= -B_1 D_U^{-1} D_U L_U^{-1} = -B_1 L_U^{-1}$$

$\square$

The entry $\left[\bar{P}_1^\top\right]_{su}$, where $s \in S$ and $u \in U$, is the probability that a random walker starting at node $u$ will be absorbed by $s$. That is because $\left[\bar{P}_1^\top\right]_{su} = \sum_{i=1}^{\infty} P_{su}(k)$ where $P_{su}(k)$ is the probability of being at node $s$ at $k$th step if you started at $u$.

**Remark B.4.** Note that $\bar{P}_1^\top = -B_1 L_U^{-1}$ is the transposed version of the linear system solved by the undirected Random Walker and equivalently the ProbWS [5, 6], i.e., $\bar{P}_1 = -\left(L_U^\top\right)^{-1} B_1^\top$. Since [5, 6] consider an undirected graph, which can be interpreted as a directed graph where each undirected edge has been replaced by a pair of directed edges in opposite directions with the same weight, the equality $\left(L_U^\top\right)^{-1} = L_U^{-1}$ holds and the transposition becomes irrelevant.

## C   Proof Theorem 4.3

In this section we prove the Theorem 4.3 of the main paper (here renamed Theorem $C.1$), which states the equivalence between the DProbWS and the Random Walker applied to directed graphs. Although it replicates step by step the same arguments presented in Theorem 4.1 of [5], we decided to include it here for reference.

**Theorem C.1.** The probability, $x_q^{s_1}$, that a random walker on a directed graph starting at node $q$ first reaches $s_1$ before reaching $s_2$ is equal to the Directed Probabilistic Watershed's probability (Definition 3 of the main paper)

$$x_q^{s_1} = P(q \sim s_1).$$

**Proof:** We write the probability of the DProbWS in terms of the inverse of $L^{[s_2]}$ (Theorem 3.4 in the main paper):

$$\Pr(q \sim s_1) = \ell_{s_1 q}^{-1,[s_2]}/\ell_{s_1 s_1}^{-1,[s_2]}. \tag{6}$$

Therefore, we only need to know the row $s_1$ of $\left(L^{[s_2]}\right)^{-1}$ to calculate for each $q$ the probability $P(q \sim s_1)$, which can be computed solving the following linear system

$$\left(L^{[s_2]}\right)^\top y = \mathbf{1}_{s_1}/\ell_{s_1 s_1}^{-1,[s_2]} \iff y = \left(\left(L^{[s_2]}\right)^\top\right)^{-1}\mathbf{1}_{s_1}/\ell_{s_1 s_1}^{-1,[s_2]} = \left(\left(L^{[s_2]}\right)^\top\right)^{-1}_{\cdot,s_1}/\ell_{s_1 s_1}^{-1,[s_2]}. \tag{7}$$

Here $\mathbf{1}_{s_1}$ denotes the column $s_1$ of the identity matrix. Note that $y$ is the vector formed by the elements in the right hand side of (6). Let us assume without loss of generality that the row corresponding to the seed $s_1$ is the first one. Thus, we can express equation (7) block-wise :

$$\begin{pmatrix} L_{s_1 s_1} & \left[B_2^\top\right]_{s_1} \\ \left[B_1^\top\right]_{s_1} & L_U^\top \end{pmatrix}\begin{pmatrix} y_{s_1} \\ y_U \end{pmatrix} = \begin{pmatrix} L_{s_1 s_1} y_{s_1} + \left[B_2^\top\right]_{s_1} y_U \\ \left[B_1^\top\right]_{s_1} y_{s_1} + L_U^\top y_U \end{pmatrix} = \begin{pmatrix} 1/\ell_{s_1 s_1}^{-1,[s_2]} \\ 0 \end{pmatrix}, \tag{8}$$

where $L_{s_1 s_1}$ is the entry $s_1 s_1$ of the Laplacian, $[B_1]_{s_1}$ and $[B_2]_{s_1}$ are the row and column $s_1$ of the Laplacian without considering the element in the diagonal and $L_U$ are the rows and columns of the unseeded vertices. Since $y_{s_1} = \Pr(q \sim s_1) = 1$, we obtain the following linear system of equations

$$L_U^\top y_U = - \left[ B_1^\top \right]_{s_1},\tag{9}$$

which is the same linear system that the Directed Random Walker solves (see Appendix B). Therefore $P(q \sim s_1) = y_q = x_q^{s_1}$ for all $q$. $\qquad\square$

## D  Proof Theorem 5.1

In this section we prove the Theorem 5.1 of the main paper (here renamed as Theorem $D.1$), which shows that in the limit case in which the Gibbs distribution has infinitely low temperature, the labeling of the DProbWS is equal to the one induced by the restriction of the DProbWS to the incoming directed forest of minimum cost, mSF, or equivalently, the incoming directed forest of maximum weight, MSF. The theorem follows the reasoning presented in Theorem 5.1 of [5].

**Theorem D.1.** Let $w_{\max} =: \max_{f \in \mathcal{F}_{\vec{s_1}}^{\vec{s_2}}} w(f)$. Given two seeds $s_1$ and $s_2$, if the entropy of the Gibbs distribution over the in-forests is minimized then

$$x_q^{s_1} = \frac{\left| \{ f \in \mathcal{F}_{s_1,q}^{s_2} \; : \; w(f) = w_{\max} \} \right|}{\left| \{ f \in \mathcal{F}_{s_1}^{s_2} \; : \; w(f) = w_{\max} \} \right|}.$$

**Proof:** The entropy of the Gibbs distribution (equation (6) of the main paper) is minimized when $\mu \to \infty$. For a fixed $\mu_0 > 0$, let us define $\mu = \alpha \cdot \mu_0$, then

$$w(e) = \exp(-\mu c(e)) = \exp(-\alpha \cdot \mu_0 c(e)) = w_0(e)^\alpha,$$

where $w_0(e) = \exp(-\mu_0 c(e))$. Since $\mu \to \infty \iff \alpha \to \infty$, we have

$$P_\alpha(q \sim s_1) := \frac{\displaystyle\sum_{f \in \mathcal{F}_{\vec{s_1},q}^{\vec{s_2}}} \prod_{e \in f} w_0(e)^\alpha}{\displaystyle\sum_{f \in \mathcal{F}_{\vec{s_1}}^{\vec{s_2}}} \prod_{e \in f} w_0(e)^\alpha} = \frac{\displaystyle\sum_{f \in \mathcal{F}_{\vec{s_1},q}^{\vec{s_2}}} w_0(f)^\alpha}{\displaystyle\sum_{f \in \mathcal{F}_{\vec{s_1}}^{\vec{s_2}}} w_0(f)^\alpha} = \frac{\displaystyle\sum_{f \in \mathcal{F}_{\vec{s_1},q}^{\vec{s_2}}} \left( \frac{w_0(f)}{w_{\max}} \right)^\alpha}{\displaystyle\sum_{f \in \mathcal{F}_{\vec{s_1}}^{\vec{s_2}}} \left( \frac{w_0(f)}{w_{\max}} \right)^\alpha} \xrightarrow[(\star)]{\alpha \to \infty} x_q^{s_1}\tag{10}$$

In $(\star)$ we used the fact that $\frac{w(f)}{w_{\max}} < 1 \iff w(f) \neq w_{\max}$. When $\alpha \to \infty$, only for the MSFs the fraction $\left( \frac{w(f)}{w_{\max}} \right)^\alpha$ does not tend to 0, but to 1. Thus, we are counting MSFs. $\qquad\square$

## E  Further experiment details

### E.1  Reference methods

**ARW method**  The method proposed in [2] is closely related to ours, due to the use of Absorbing Random Walks (ARW). Conceptually, this method adds an absorbing meta node to the original graph, which is connected to every node in by an edge with certain weight $\bar{w}$.[1] The algorithm computes the random walker expected accumulated number of visits to the seeds starting at a query node before being absorbed by the meta node. The expected accumulated number of visits provides a notion of affinity between the nodes. The algorithm assigns the label of the seed that maximizes the expected accumulated number of visits from the unlabeled query node.

**LLUD method**  The method proposed in [16] also uses the random walker. The LLUD method considers an optimization problem over the space of classification functions $\mathcal{H}(V)$, which assigns a real value $f(v)$ to each vertex $v \in V$. The optimization problem is the following

$$\arg \min_{f \in \mathcal{H}(V)} \{ \Omega(f) + \gamma ||f - y|| \}\tag{11}$$

---

[1]The weight $\bar{w}$ is implicitly determined by a parameter $\alpha$ in the algorithm, which we set equal to 0.1 for all our experiments. See [2] for more details.

where $y$ denotes the function in $\mathcal{H}(V)$ defined by $y(v) = 1$ or $-1$ if vertex $v$ has been labeled as positive or negative respectively, and 0 if it is unlabeled. The functional $\Omega$ is defined as

$$\Omega(f) \coloneqq \frac{1}{2} \sum_{(u,v) \in E} \pi(u) P_{uv} \left( \frac{f(u)}{\sqrt{\pi(u)}} - \frac{f(v)}{\sqrt{\pi(v)}} \right), \tag{12}$$

where $\pi$ is the stationary distribution of the random walker, which is assumed to be unique independently of the starting point. [2] The functional $\Omega$ forces the classification function to be smooth, while the second term in (11) forces the function to preserve the label of the seeds. The balance between this two terms is regulated by the parameter $\gamma$.[3]

**GTG method**   The method in [4] interprets the classification of the nodes as a transductive game where each player (node) can choose a strategy among a set of strategies (labels). The proposed transduction game always has a Nash equilibrium which will define the final labeling. Partial payoffs between two nodes are defined based on the weight of the edge connecting these nodes. The total payoff of a node is the sum of its partial payoffs. The Nash equilibrium is computed iteratively till convergence.

### E.2   Datasets

**Digits**   The *Digits* dataset[4] [3, 14] consists of $8 \times 8$ pixel images of digits. The dataset contains a total of 1797 images divided in 10 classes corresponding to the different digits. We use this dataset to construct a kNN graph. The kNN graph is formed by 1797 nodes and 8985 edges.

**20Newsgroups**   The *20Newsgroups* dataset[5] [3, 9] is a collection of 11314 newsgroup documents, partitioned across 20 different newsgroups. We only use the train data. We define a kNN graph out of it. The kNN graph contains 11314 nodes and 56570 edges.

**Email-EU**   The *Email-EU* dataset[6] [10, 11, 15] is a directed unweighted graph that was generated using email data from a large European research institution. An edge $(u, v)$ is present in the graph if person $u$ sent an email to person $v$. There is a total of 1005 nodes and 25571 edges. It has a total of 42 classes representing the departments at the research institute.

**Cora**   The *Cora* dataset[7] [12] is a directed graph where each node represents a scientific publication. An edge $(u, v)$ is present in the graph if paper $u$ cites $v$. There is a total of 2708 nodes and 5429 edges. It has a total of 7 classes representing the topics of the publications.

**CiteseerX**   The *CiteseerX* dataset[8] [13] is a directed graph where each node represents a scientific publication. An edge $(u, v)$ is present in the graph if paper $u$ cites $v$. There is a total of 3264 nodes and 4536 edges. It has a total of 6 classes representing the topics of the publications.

### E.3   k-Nearest Neighbor graph construction

The *Digits* and *20Newsgroups* datasets are not graph datasets. To process them with the DProbWS, we construct k-Nearest Neighbor (kNN) graphs out of them. Directed edges of the kNN graphs are obtained as follows: an edge from node $u$ to node $v$ is present if and only if $v$ is among the k nearest neighbors of $u$. In our experiments we set $k = 5$ as in [2].

---

[2]In practice, the uniqueness is ensured thanks to the use of the teleporting random walk.

[3]In the algorithm, the parameter $\gamma$ is implicitly determined by another parameter $\mu$, which we set equal to 0.9 for all our experiments. See [16] for more details.

[4]`https://scikit-learn.org/stable/modules/generated/sklearn.datasets.load_digits.html`

[5]`https://scikit-learn.org/stable/modules/generated/sklearn.datasets.fetch_20newsgroups.html`

[6]`http://snap.stanford.edu/data/email-Eu-core.html`

[7]`https://web.archive.org/web/20151007064508/http://linqs.cs.umd.edu/projects/projects/lbc/`

[8]`https://networkrepository.com/citeseer.php`

To generate the kNN graph first we need to embed the data points into a metric space. In the case of the *Digits* dataset, since they are images, we just flatten the 8x8 images into a vector of 64 dimensions. The *20Newsgroups* is a text dataset. Via the `TfidfVectorizer` class implemented in `scikit-learn` in Python[9], we embed the datapoints into $\mathbb{R}^{130107}$. `TfidfVectorizer` maps a collection of raw documents to a matrix of TF-IDF features, where TF-IDF stands for "Term Frequency - Inverse Document Frequency". TF-IDF is a statistic that aims to better define how important a word is for a document, while also taking into account the relation to other documents from the same corpus [1].

Once the data points have been embedded, the weight of the assigned edge $(u, v)$ is given by $w_{uv} = \exp(-||f(u) - f(v)||)$, where $f(u)$ is the image of the node $u$ in the embedding space. Note that the structure of the graph and the weights depend on the distances between the nodes in the embedding space. If the representation of the data in the embedding space is poor, so will be the labeling.

### E.4 Hardware

The hardware we used is a machine with Intel Xeon E5-2650V3 CPU.

### E.5 Teleporting random walker

As pointed out in Remark 3.1, if a node cannot reach any seed via a directed path, the DProbWS method is not able to infer the label. Such nodes are known as zero-knowledge nodes. From the tree perspective view, the zero knowledge nodes do not belong to any in-tree rooted at the seeds and therefore the DProbWS probabilities are all equal to zero.

Due to the high sparsity of the citation networks (Cora and CiteseerX), many of the nodes are zero-knowledge nodes and therefore its label can not be inferred. Inspired by [16], we remedy this by replacing the natural random walker by the so-called teleporting random walker (TRW). In the TRW setting, a random walker jumps uniformly at random to any node with probability $\eta$, and with probability $(1 - \eta)$ takes a step of the natural random walker. This ensures that any node can reach any other node. Let $P$ and $P_{TRW}$ be the transition probability matrices of the random walker and the teleporting random walker. Formally, they are related as follows

$$P_{TRW} = (1 - \eta)P + \frac{\eta}{n - 1} \left( \mathbb{1}\mathbb{1}^\top - I \right), \tag{13}$$

where $\mathbb{1}$ is the column vector full of ones, $I$ is the identity matrix of appropriate size and $n$ is the number of nodes in the graph.

Since both ARW [2] and DProbWS methods are based on the random walker, we can implement the variants ARWtrw and DprobWStrw, which make use of the TRW. We set the $\eta$ value equal to $10^{-6}$ for all datasets except for the *Digits* dataset, where $\eta = 10^{-2}$.

In practice, the TRW approach is equivalent to add to every node out-going edges towards the rest of the nodes. By adding these out-going edges to each node the graph becomes a complete graph. Hence, we can not exploit the efficient sparse linear solvers. The next section proposes an efficient computation of the DProbWStrw probabilities, that permits to use sparse linear solvers at the expense of solving an extra linear system. The results presented in the Experiments section for the TRW variants have been been computed without this speed-up by inverting the dense Laplacians.

### E.5.1 Efficient computation of the DProbWStrw probabilities

In this section we will show that in the DProbWStrw variant, to use a TRW with self-loops is equivalent to use the TRW without self-loops. Thanks to this equivalence, we can exploit the Sherman-Morrison formula (Lemma $E.2$) which permits to take advantage of the sparsity of the graph.

In Theorem 4.2 we have proven the equivalence between the random walker and the in-forest approaches. Thanks to this equivalence we can easily show that self-loops do not play a role in the

---

[9] `https://scikit-learn.org/stable/modules/generated/sklearn.feature_extraction.text.TfidfVectorizer.html`

DProbWS probabilities. Since any in-forest does not contain cycles, and therefore self-loops, the set of in-forests of a graph remains the same if we add self-loops to the graph. Therefore, the total weight of the set of in-forests is not modified by adding self-loops to the graph. Consequently, the DProbWS probability is unchanged (see Definition 3). In light of Theorem 4.2, this implies that the random walker absorption probabilities of the seeds are also independent of the presence of self-loops. Intuitively, we can argue that in the long run the number of steps the random walker remains immobile at a node (number of steps that the random walker traverses a self-loop) is irrelevant to the seed absorption probability.

Formally, the graph with self-loops and without self-loops define the same Laplacian matrix. Let $A$ be the adjacency matrix of the graph without self-loops. If we add self-loops to the graph, the adjacency matrix of the graph with self-loops will be $\hat{A} = A + d$, where $d$ is a diagonal matrix indicating the weights of the self-loops edges. Therefore,

$$L = D - A^\top = \underbrace{D + d}_{\hat{D}} - \underbrace{(A^\top + d)}_{\hat{A}^\top} = \hat{D} - \hat{A}^\top = \hat{L},$$

i.e., the Laplacian of the graph without self-loops, $L$, and with self-loops, $\hat{L}$, are the same. As a consequence of Theorem 3.5, the DProbWS probabilities are also equal for both graphs.

Note that, although the Laplacians are unaffected by adding self-loops, the random walker transition probabilities at each node will differ by adding a self-loop. However, a random walker defined on a graph with self-loops will determine the same absorption probabilities as a random walker on the same graph without the self-loops. This equivalence holds because the transition probabilities of the graph with self-loops, at each node, conditioned to not traverse the self-loop of the node are equal to the transition probabilities in the graph without the self-loops.

The following lemma proposes a variant of the TRW with self-loops which defines the same DProbWS probabilities as the TRW without self-loops described in the previous section (13).

**Lemma E.1.** The TRW without self-loops proposed in (13) determines the same DProbWS probabilities as the TRW with self-loops defined by the following transition probability matrix

$$\hat{P}_{TRW} = (1 - \hat{\eta})P + \hat{\eta}\frac{1}{n}\mathbb{1}\mathbb{1}^\top, \tag{14}$$

where $\hat{\eta} = \frac{n\eta}{\eta + n - 1}$.

**Proof:** As argued in the previous paragraphs, we just need to show that the transition probabilities of the TRW with self-loops at each node conditioned to not traverse the self-loop are equal to the transition probabilities in the graph without the self-loops. The probability of traversing a self-loop in the TRW with self-loops is equal to

$$\hat{P}_{TRW}(\text{self-loop}) = \frac{\hat{\eta}}{n}.$$

Consequently, the transition probability of the TRW with self-loops from node $i$ to node $j$ conditioned to not traverse the self-loop is given by

$$\hat{P}_{TRW}(j|i, \neg\text{self-loop}) = \begin{cases} \frac{\hat{P}_{TRW}(j|i)}{1 - \hat{P}_{TRW}(\text{self-loop})} = \frac{\hat{P}_{TRW}(j|i)}{1 - \frac{\hat{\eta}}{n}} & \text{if } i \neq j \\ 0 & \text{if } i = j \end{cases}.$$

Hence, the transition probability of the TRW with self-loops conditioned to not traverse any self-loop will be equal to

$$\hat{P}_{TRW, \neg\text{self-loop}} = \frac{(1 - \hat{\eta})}{1 - \frac{\hat{\eta}}{n}}P + \frac{\hat{\eta}}{1 - \frac{\hat{\eta}}{n}}\frac{1}{n}\left(\mathbb{1}\mathbb{1}^\top - I\right).$$

Substituting $\hat{\eta}$ by $\frac{n\eta}{\eta + n - 1}$ we obtain

$$\hat{P}_{TRW, \neg\text{self-loop}} = (1 - \eta)P + \frac{\eta}{n - 1}\left(\mathbb{1}\mathbb{1}^\top - I\right) = P_{TRW},$$

i.e., we retrieve the transition probability matrix of the TRW without self-loops (13). □

Next, we will express the linear system that the DProbWStrw solves in terms of the $\hat{P}_{TRW}$ matrix, instead of the Laplacian. The DProbWS method solves the following linear system (Theorem 4.2) for each seed $s$

$$L_U^\top x_U^s = -\left[B_1^\top\right]_s.\tag{15}$$

Thanks to the relation between the transition probability matrix and the Laplacian exposed in (4), the linear system (15) is equivalent to

$$\left(I - P_U^\top\right) x_U^s = -\left[\hat{B}_1^\top\right]_s,\tag{16}$$

where $I$ is the identity matrix with the appropriate size and $\hat{B}_1^\top = D_U^{-1} B_1^\top$, with $D$ the diagonal matrix whose entries are the out-degree of each node. [10]

If we use the TRW with self-loops, as a consequence of (14), equation (16) becomes

$$\left((1 - \hat{\eta})\left(I - P_U^\top\right) + \frac{\hat{\eta}}{n}\mathbb{1}\mathbb{1}^\top\right) x_U^s = -(1 - \hat{\eta})\left[\hat{B}_1^\top\right]_s - \frac{\hat{\eta}}{n}\mathbb{1}.\tag{17}$$

or alternatively

$$\left(\left(I - P_U^\top\right) + \frac{\hat{\eta}}{n(1 - \hat{\eta})}\mathbb{1}\mathbb{1}^\top\right) x_U^s = -\left[\hat{B}_1^\top\right]_s - \frac{\hat{\eta}}{n(1 - \hat{\eta})}\mathbb{1}.\tag{18}$$

The difference between the matrices of the linear systems in (16) and (18) is equal to a 1-rank matrix

$$\left(I - P_U^\top\right) - \left(\left(I - P_U^\top\right) + \frac{\hat{\eta}}{n(1 - \hat{\eta})}\mathbb{1}\mathbb{1}^\top\right) = -\frac{\hat{\eta}}{n(1 - \hat{\eta})}\mathbb{1}\mathbb{1}^\top.$$

Therefore, we can apply the Sherman-Morrison formula [7].

**Lemma E.2** (Sherman-Morrison formula [7]). Given an invertible matrix $A \in \mathbb{R}^{m \times m}$ and $u, v \in \mathbb{R}^m$ we have

$$\left(A + uv^\top\right)^{-1} = A^{-1} - \frac{A^{-1}uv^\top A^{-1}}{1 + v^\top A^{-1}u}.$$

if and only if $v^\top A^{-1} u \neq 1$.

Let $N = I - P_U^\top$, then we can apply Lemma $E.2$ to (18) and obtain

$$
\begin{aligned}
x_U^s &= \left(\left(I - P_U^\top\right) + \frac{\hat{\eta}}{n(1 - \hat{\eta})}\mathbb{1}\mathbb{1}^\top\right)^{-1}\left(-\left[\hat{B}_1^\top\right]_s - \frac{\hat{\eta}}{n(1 - \hat{\eta})}\mathbb{1}\right) \\
&= \left(N + \frac{\hat{\eta}}{n(1 - \hat{\eta})}\mathbb{1}\mathbb{1}^\top\right)^{-1}\left(-\left[\hat{B}_1^\top\right]_s - \frac{\hat{\eta}}{n(1 - \hat{\eta})}\mathbb{1}\right) \\
&\underbrace{=}_{\text{Lemma } E.2} \left(N^{-1} - \frac{\hat{\eta}}{n(1 - \hat{\eta})}\frac{N^{-1}\mathbb{1}\mathbb{1}^\top N^{-1}}{1 + \frac{\hat{\eta}}{n(1-\hat{\eta})}\mathbb{1}^\top N^{-1}\mathbb{1}}\right)\left(-\left[\hat{B}_1^\top\right]_s - \frac{\hat{\eta}}{n(1 - \hat{\eta})}\mathbb{1}\right) \\
&= -N^{-1}\left[\hat{B}_1^\top\right]_s - \frac{\hat{\eta}}{n(1 - \hat{\eta})}N^{-1}\mathbb{1} \\
&\quad + \frac{\hat{\eta}}{n(1 - \hat{\eta})}\frac{N^{-1}\mathbb{1}\mathbb{1}^\top N^{-1}\left[\hat{B}_1^\top\right]_s}{1 + \frac{\hat{\eta}}{n(1-\hat{\eta})}\mathbb{1}^\top N^{-1}\mathbb{1}} + \left(\frac{\hat{\eta}}{n(1 - \hat{\eta})}\right)^2\frac{N^{-1}\mathbb{1}\mathbb{1}^\top N^{-1}\mathbb{1}}{1 + \frac{\hat{\eta}}{n(1-\hat{\eta})}\mathbb{1}^\top N^{-1}\mathbb{1}}
\end{aligned}\tag{19}
$$

---

[10]We assume that the out-degree is distinct of zero for all nodes in the set of unlabeled nodes $U$.

Equation (19) shows that $x_U^s$ can easily be computed via (18), if we compute $N^{-1}\left[\hat{B}_1^\top\right]_s$ and $y := N^{-1}\mathbb{1}$. Computing the term $N^{-1}\left[\hat{B}_1^\top\right]_s = I - P_U^\top\left[\hat{B}_1^\top\right]_s$ is equivalent to solve the linear system in (15). Hence, by exploing the Shermann-Morrison formula (Lemma $E.2$), we just need to solve the extra linear system $Ny = \mathbb{1}$, which costs much less effort than solving a dense linear system per seed.