# OpenReview forum: "Directed Probabilistic Watershed"
_NeurIPS.cc/2021/Conference — NeurIPS 2021 Poster_

### Official Review · Reviewer_CQ5E · 2021-07-10

**Rating:** 7
**Confidence:** 4

**Summary:**

The paper proposes a semi-supervised learning algorithm for segmenting directed graphs - obtaining extensions to several previously proposed methods for segmenting undirected graphs. The authors' build on  the 2019 Probabilistic Watershed (ProbWS) paper [12], a graph segmentation algorithm, to construct the directed graph extension. The author's use the Matrix Tree Theorem (MTT), obtaining a Gibbs distribution characterization that they prove is equivalent to the Directed Random Walker (DRW) image segmentation algorithm [15], in analogy to the undirected case studied in [12].  analytical computations of probabilities. They then obtain a directed version of the Power Watershed (PW) graph partitioning algorithm [6]. A numerical study is performed to illustrate and compare the directed ProbWS approach.







**Ethical Concerns:**

No ethical concerns.

**Limitations And Societal Impact:**

No negative implications.

**Main Review:**

The paper is well written, original, and makes significant contribution. The one weakness is the incompleteness of numerical results and comparisons.

Novelty:
The author's extension of ProbWS [12] and PW [6]  to directed graphs is interesting and non-trivial.  Using the directed version of the Matrix Tree Theorem they provide a way to analytically compute the directed probabilistic watershed probabilities, specified by Gibbs probability distributions, that specifiy the probability that an (unlabelled) graph node is assigned to a particular seed (labelled node) in an ensemble of incoming directed spanning forests rooted at the seed. This results in Thm 3.5, characterizing the probabilities exactly, and an equivalence of ProbWS to  DRW.


Scholarship:
The paper does an excellent job relating the contribution to prior work, which is extensively discussed and cited in the paper. They make it very clear what their contribution is and what innovation on the theory is needed to develop their extension.


Experimental completeness: The experimental section is brief - most of the paper centers on the theoretical contributions. The authors show favorable comparisons of the directed ProbWS to another algorithm ARW [9] for 3 different data sets (Digitsm 20Newsgroups, Email-EU). More extensive numerical illustrations would help to establish the practical significance of the paper.  This would include: comparisons to other methods proposed for directed graph segmentation, e.g., spectral methods [Fagiolo, PR-E 2007], graph cuts [Meila and Pentney, SIAM Data Mining 2007] and conductance methods [Yin et al, KDD 2017]; exploration of the  influence of the choice of the fraction of nodes (r) used as seeds; influence of the local topology of these seeds; and scalability of the computation to larger networks.


Quality of writing:
The paper is very well-written.


Authors' statements on the questionnaire:
Question 3 not accurate - there is no code provided in the paper as far as we have seen. The hardware is mentioned in the supplementary material. Some details on implementation are mentioned in the supplementary material as well, including some of the hyperparameters.  However, it is not explained why their particular value of r was chosen to create the directed graphs.


**Time Spent Reviewing:**

2-3 hours

---

> ### Author Response · Authors · 2021-08-10
> **Specific comments: Reviewer CQ5E**
>
> We appreciate the positive review and the very helpful and detailed comments. We appreciate that you found that our work is “well written, original, and makes significant contribution”. We will address the concerns in the revised manuscript.
>
> 1) Experimental completeness: As stated in the paper, the focus of our paper was to emphasize the  the extension of the Probabilistic Watershed to the directed case, instead of proving the practical superiority of the method. Nonetheless, we addressed part of your suggestions in the response directed to all reviewers (Experimental completeness). We thank the reviewer for the suggestions and for pointing us to such interesting literature. A fair comparison to those methods seems difficult since they are unsupervised methods while ours is semi-supervised.
>
> 2) Code missing: Though we attached the code in the supplemental material, we plan to describe the algorithm more explicitly in the revised version and we will also make our code publicly available upon acceptance. The values such $k$ and $r$ were chosen based on the experiments carried in [9], and we will mention it in the revised manuscript.

---

> ### Comment · Reviewer_CQ5E · 2021-08-26
> **I maintain my score**
>
> I still think this paper is in the acceptance range. The authors have done a good job quantifying  the comparative performance in their response to the other reviewers.

---

### Official Review · Reviewer_L2ar · 2021-07-14

**Rating:** 7
**Confidence:** 4

**Summary:**

The paper introduces an extension of the Probabilistic Watershed method in semi-supervised learning from undirected graphs to directed graphs. The paper makes the connection between the Directed Probabilistic Watershed and the Directed Random Walker, which is an extension of the Ransom Walker to directed graphs and is also introduced in this paper.

**Limitations And Societal Impact:**

The authors have addressed the societal impact, but not the limitations.

**Main Review:**

The paper brings forth a theoretical contribution in the form of the Directed Probabilistic Watershed and Directed Random Walker methods. The main results of the paper are the computation of the probabilities that any query node is connected to any of the labeled nodes.
However, these contributions are purely theoretical and the authors make no effort in being more accessible to the researchers that might just want to use the method for their semi-supervised learning or image segmentation application. To this end, a precise description of the Directed Probabilistic Watershed or Directed Random Walker algorithm is missing from the paper. This limitation makes the paper only accessible to theoretical researchers and limits its practical applicability.
In contrast, the Random Walker is a segmentation algorithm with an exact description that can be used for labeling undirected graphs.
The experiments are limited and not described properly, therefore not reproducible. It is not clear how the kNN graph was obtained, what are the nodes, and what the edge weights are.

**Time Spent Reviewing:**

2

---

> ### Author Response · Authors · 2021-08-10
> **Specific comments: Reviewer L2ar**
>
> We appreciate the time and effort you put into the comments.
>
> 1) Reproducibility: As stated in line 289, we detail all parameters of our experiments and the graph structure in the supplemental material. In order to ease the reproducibility and make it accessible to the practitioners, we will make our code publicly available upon acceptance
>
> 2) Description algorithm: The Directed Probabilistic Watershed algorithm is identical to the undirected Random Walker algorithm [9,15] with the difference that it uses the transposed of the out-Laplacian (Definition 2) instead of the common Laplacian matrix. For the revised manuscript, we will give a precise description of the algorithm (which we briefly sketch below). We thank the reviewer for bringing this to our attention.
>
> ```
> INPUT: (Graph, seeds}
> OUTPUT: C #classification of the nodes
>
>     #Define matrices
>     L=out_Laplacian(Graph)
>     L_U=obtain_L_U(L,seeds)# squared submatrix of L indexed by the unlabeled nodes
>     B=obtain_B(L,seeds)# submatrix of L: rows indexed by unlabeled nodes; columns by seeds
>
>     #Compute absorbing probabilities P
>     P=solve(L_U.transpose,B)#Solves the linear system (L_U.transpose)*P=B
>
>     #Obtain classification
>     C=argmax(P,1) #Select seed with highest absorbing probability.
>     return C
> ```

---

> > ### Comment · Reviewer_L2ar · 2021-08-25
> > **I maintain my rating**
> >
> > The authors have addressed my issues with the unclear algorithm description.
> > However, the experiments are not clearly described even in the supplementary material. It is not clear how the data was divided into training and testing datasets and whether the kNN graph was constructed from the training set or all the data.
> > Because of these, I maintain my rating.

---

> > > ### Author Response · Authors · 2021-08-25
> > > **Split data**
> > >
> > > We thank you for your reply. We lament that the experiment details exposed in the main paper and the supplemental material did not completely clear out all your concerns about how these experiments have been carried out.
> > >
> > > Regarding the splitting of the data, we do not quite understand what do you refer to. The DProbWS is a semi-supervised learning algorithm, which implies that all the data (labeled and unlabeled) is used for learning. From this point of view, we could consider the whole dataset as training set. Consequently, the data has not been split into training and test sets.
> > >
> > > In case that by training and testing datasets, you refer to how the labeled data was assigned, we did the following: we sampled uniformly a certain fraction $r$ of nodes from each class of the data and set them as seed nodes. We sampled the labeled nodes for different $r$ values between 0.1 and 0.9 (line 291).
> > >
> > > For the construction of the kNN-graph, we used the whole dataset. This construction is independent of the labeled and unlabeled state of the nodes. First, we embedded the data points into a metric space as explained in the supplemental material (see lines 143-150). Once all the data is embedded, we define the kNN-graph. The weight of an edge e=(u,v) is set equal to $w_{uv}=\exp(d(f(u),f(v))$, where $d(·,·)$ is the distance function of the metric space and $f(u)$ is the image of the node $u$ in the embedding space.
> > >
> > > We hope that this addressed your concern. In the revised manuscript, we will make explicit these points to avoid confusion.

---

> > > > ### Comment · Reviewer_L2ar · 2021-08-25
> > > > **Still not quite right**
> > > >
> > > > Even a semisupervised algorithm is supposed to learn from some data and be used on other data later on.
> > > > For example, a doctor might want to use your algorithm to predict some clinical variables, and he will use a trained model (trained in a semisupervised way) on data that has not been used for training. One cannot expect the doctor to retrain the model as soon as new data is presented. Thus the train/test split is relevant even in semi-supervised settings.
> > > > But more importantly, the number of labeled observations changes in the accuracy evaluation. Without using a set-aside test set, your accuracy is a mixture between the accuracy on the labeled data and the accuracy on the unlabeled data. This mixture varies with the parameter r. If we take r to the extreme and make it r=1, your reported accuracy is the training accuracy, which is not what we want. At least you could report the accuracy on the unlabeled data, but you are not doing that, or it is not clear from the text that you are reporting that.

---

> > > > > ### Author Response · Authors · 2021-08-26
> > > > > **Transductive learning**
> > > > >
> > > > > The DProbWS semi-supervised learning algorithm is an instance of **transductive** graph-based semi-supervised learning. In contrast to inductive semi-supervised learning, where a model is trained on labeled and unlabeled data in order to learn general rules that can be applied to unseen data, transductive learning techniques have access to the whole data (labeled and unlabeled) and only aim to infer the labels of the unlabeled data by exploiting its relation to the labeled data.
> > > > >
> > > > > The methods we compare against are transductive, too, as is the undirected Random Walker algorithm which is very popular, for instance on image data [15,f,g]. The practical advantage of transductive learning is that given a large dataset, the user only has to provide few labels to get a good accuracy on all nodes. Consider, as an example, the digits dataset, where given only labels for 10% of the nodes, more than 95% get correctly classified by DProbWS.
> > > > >
> > > > > We reported the accuracy on labeled and unlabeled data as it is done in the literature [9,11]. So the accuracies are lower-bounded by $r$. The difference between the obtained accuracy and $r$ can be seen as the successful transductive learning.
> > > > >
> > > > > [f] "Multilabel random walker image segmentation using prior models” Grady, CVPR, 2005
> > > > >
> > > > > [g]"End-to-End Learned Random Walkerfor Seeded Image Segmentation”, Cerrone et al. CVPR, 2019

---

> > > > > > ### Comment · Reviewer_L2ar · 2021-08-27
> > > > > > **OK**
> > > > > >
> > > > > > Ok, if everybody else is doing it.
> > > > > > But in [11] the authors explicitly mention what accuracy measure they used, while your paper did not.

---

> > > > > > > ### Author Response · Authors · 2021-08-30
> > > > > > > **Measure definition**
> > > > > > >
> > > > > > > We agree that we did not explicitly define the accuracy measure we used (correctly labeled nodes divided by the total number of nodes). We will amend this in the revised version.

---

> > > > > > > > ### Comment · Reviewer_L2ar · 2021-08-31
> > > > > > > > **RE: Measure definition**
> > > > > > > >
> > > > > > > > In that case, I update my rating to "Accept".

---

### Official Review · Reviewer_ctCN · 2021-07-17

**Rating:** 4
**Confidence:** 4

**Summary:**

This paper proposes an extension of a previous graph-based method for semi-supervised learning. Specifically, the proposed method builds upon the probabilistic watershed algorithm, which is a transductive graph-based algorithm for semi-supervised classification, based on building "separating forests" (i.e., in which each tree contains a single labeled node) and defining a Gibbs distribution on this set of forests.  The main contribution of the paper is a generalization of this method to directed graphs, and showing a couple of theoretical results related to the proposed generalization.

**Ethical Concerns:**

None.


**Limitations And Societal Impact:**

They have.


**Main Review:**

This is an interesting paper, and the proposed contribution is certainly not trivial and generalizes the class of methods upon which it builds. Furthermore, the theoretical results, namely the equivalence with the random walker method is interesting and completes the proposed generalization.

On the negative side, I think the contribution of the paper is rather marginal and the experimental evaluation of the proposed method is quite limited. The motivation of why it is at all relevant to use directed graphs in graph-based semi-supervised classification is only mentioned in the experiments section, where the authors use kNN graphs, which are not guaranteed to be symmetric. The only comparison is with another method for directed graphs, but this begs the obvious question: What would be the results of symmetrizing the graph and then using the undirected probabilistic watershed algorithm? This comparison, if advantageous for the proposed method, would give some experimental support to the relevance of what is proposed in the paper.

The authors do claim that "although our work has been motivated from a theoretical perspective, we run an illustrative experiment", but this is not a good enough excuse. These graph-based semi-supervised classification methods are very heuristic, in the sense that they lack any decision-theoretic support or any formal (statistical learning theoretic) guarantees. They are based on ideas that seem reasonable, but that is not enough to justify their relevance. Consequently, for this type of method, "the proof is in the pudding", thus experimental results are all there is to justify their relevance and applicability, and the only way to justify the relevance of some new contribution to this class of methods.

True, the methods are mathematically sophisticated and interesting, and the paper even includes a few interesting theorems. However, from a machine learning perspective, these theorems are somewhat "internal", in the following sense: they establish properties of the methods, and equivalences with related methods, but they say nothing about the performance of the method.



**Time Spent Reviewing:**

4

---

> ### Author Response · Authors · 2021-08-10
> **Specific comments: Reviewer ctCN**
>
> We thank the reviewer ctCN for the time and effort invested to read our work. We are pleased that you found that our work is interesting and non-trivial. Your concerns have been addressed in the response directed to all reviewers (Experimental completeness and Symmetrization of the kNN graph).

---

### Official Review · Reviewer_1WWq · 2021-07-20

**Rating:** 6
**Confidence:** 4

**Summary:**

Suppose we have a weighted directed graph with 2 (or more) nodes designated as "seeds" (call them 0 and 1), and we want to assign a "score" to each node depending on its proximity to seed 0 relative to seed 1. This is a sub-routine that can be used for some semi-supervised learning tasks: for example, think of the nodes as images of cats and dogs, and the two seeds as one image of each type whose classification is known; the goal is to classify the remaining images.

This paper proposes a natural method for assigning a "score" to each node in the above setup. The method is based on sampling a random forest from a certain Gibbs measure. This is proven to be equivalent to the following more intuitive interpretation: imagine starting a random walk at some node q, where the transition probabilities depend in a natural way on the directed edge weights; the "score" assigned to node q is the probability that this random walk reaches seed 1 before reaching seed 0. It is shown that these probabilities can be computed in closed form using the directed matrix tree theorem, so there is no need to actually simulate the random walk. All of this generalizes to the case where there are more than 2 seeds, in which case each node gets assigned a probability distribution over the seeds.

In the case of an undirected graph, the results of this paper were already known and called the "Probabilistic Watershed algorithm" (NeurIPS 2019). This paper provides the natural generalization to directed graphs.

The paper includes some experiments on three different datasets, including classifying images of digits 0-9. To construct the graph, each image is connected to its 5 nearest neighbors (in some metric), with a weight depending on the distance. A fraction r of the nodes are chosen at random to have their labels revealed (these are the "seeds"). The experiments show that the new method seems to perform a bit better than the prior work [9], particularly on the digits dataset.

**Limitations And Societal Impact:**

Ok

**Main Review:**

The results of this paper are a natural extension of known results for the undirected case. The results do not seem too difficult to prove, once one is familiar with the undirected case. Still, since this is a fairly fundamental problem, writing down the details of this seems like an important service to the community. Overall, I would be open to seeing the paper accepted if there is space in the program. However, the level of novelty is arguably a bit borderline.

Once the directed graph and seeds are given, the method is quite principled and feels like the "right" way to approach the problem. However, the procedure by which the graph is built from the dataset (using k-nearest neighbors) feels more "ad hoc". In particular, the original dataset doesn't seem to contain "directed" information, so transforming it into a directed graph feels a bit artificial. I wonder if one could do just as well by instead building an undirected graph.

It is interesting that the new method seems to significantly outperform [9] on the digits dataset. The ARW method from [9] appears similar at first glance, as it is also based on random walks (but seems slightly less principled than the current paper). It would improve the paper if the authors could explain why we see this discrepancy, i.e., what exactly is going wrong with the ARW method and what makes the new method better?

Specific comments:
pg 1: The weight of a forest hasn't been defined yet. It might be good to clarify that this means the product of the weights of the edges.
pg 3: The definitions here are a bit confusing because you formally define a spanning in-tree (Def 1) but then use (non-spanning) in-trees to define a "2-in-trees spanning forest".
pg 8: It would help to define "accuracy", even if informally.

**Time Spent Reviewing:**

4

---

> ### Author Response · Authors · 2021-08-10
> **Specific comments: Reviewer 1WWq**
>
> We thank you for the time and effort invested and the valuable questions. We appreciate that you found that our work is “an important service to the community”.
>
> 1) Choice kNN-graph: We decided to use kNN-graphs, with k=5, based on the experiments made in [9] by De et al. The aim of our work is to show that the results presented in the Probabilistic Watershed paper can also be extended to the directed case, and therefore our focus is fundamentally theoretical. We included the experiments section as an illustrative example. In such case, the kNN-graphs were an straightforward way of defining directed graphs. We also included the non-kNN graph “Email-EU” dataset, which is a directed unweighted graph that was generated using email data from a large European research institution. Furthermore, in the response to all reviewers (experimental completeness) you may find the accuracies obtained for two extra datasets (Cora and CiteseerX), which will be included in the final manuscript.
>
> 2) DProbWS better than ARW on the digits dataset: Indeed, both methods are principled on the random walker. The ARW algorithm measures the affinity between the nodes and the seeds based on the cumulated expected number of visits from the query nodes to the seeds. To compute these affinities, the ARW method uses the transposed adjacency matrix of the graph. Therefore, in the ARW method the actual underlying graph being used is the graph with all edges reversed. The original ARW paper [9] considers that seed nodes are usually source nodes (nodes with in-degree equal to 0). If a seed is a source, no directed path can reach those nodes and therefore the affinities to the sources is equal to 0. If one considers the reversed graph (all edge directions reversed) the sources become sinks, which justifies the transposition of the adjacency matrix. We implemented the ARW algorithm as described in the original paper by De et al., i.e using the transposed adjacency matrix. After the inquiry, we have tested different possibilities to discern why the ARW performed worse. We noticed that without reversing the edges the accuracies increased until it reached that of DProbWS. Therefore, reversing the edges is not beneficial in this case. In general, we noticed that for the other datasets was also harmful. Thus, we have decided not to transpose the adjacency matrix, as suggested by [9], and update the results for the revised manuscript. The updated results can be found in the results table of the response directed to all reviewers (Experimental completeness).
>
> 3) Specific comments: We will clarify and make explicit the pertinent definitions in order to ease the readability and avoid confusion. We are grateful for bringing this to our attention.

---

> > ### Comment · Reviewer_1WWq · 2021-08-31
> > **Response**
> >
> > I have read the rebuttal and maintain my score. The authors' response to point #2 demystifies things somewhat and is perhaps worth including in the final paper.

---

### Official Review · Reviewer_j9jR · 2021-07-21

**Rating:** 6
**Confidence:** 1

**Summary:**

This is a very interesting manuscript and an enjoyable read. Essentially, it extends random walks on graphs which perform probabilistic watersheds to random walks on directed graphs which perform directed probabilistic watershed. This is not a trivial extension and is hence a significant contribution - if correct. The caveat is necessary since it will be difficult for any conference reviewer to find the time needed to verify the result.

**Ethical Concerns:**

This is some woke stuff which is irrelevant for this manuscript.

Or rather, since the original Metropolis-Hastings MCMC sampling algorithm was connected to research on atomic bombs, perhaps this paper should be banned?

**Limitations And Societal Impact:**

The manuscript does fundamental work and so this question does not apply.

**Main Review:**

The manuscript is well written but is not easy to read since the proofs are difficult. In particular, I cannot yet verify if Theorems 4.1 and 4.3 are correct. And these are the main theorems in the manuscript.

Questions:

1. Since the "directed" Laplacian is not symmetric, notions of non-negative definiteness do not apply (and if accepted, please fix the language since the concept itself is meaningless for non-symmetric matrices). Consequently, the authors pursue a directed random walk approach instead of the objective function approach. How can you then have a Gibbs distribution if there's no "energy function" since the two are joined at the hip? Unless the distribution is not Gibbsian due to the directedness of the graph? Please clarify. Perhaps you meant the stationary distribution.
2. How is your work related to work on sampling directed graphs with random walks (Ribeiro et al.)
3. This is a tangential question. There are strong relationships between the determinants in MTT and eigenvectors. Since your matrices are not symmetric, eigenvectors if they exist can be complex. Have these relationships been worked out?
4. Can everything be done for out-forests rather than in-forests? Seems like that would be the case.




**Time Spent Reviewing:**

3 hours

---

> ### Author Response · Authors · 2021-08-10
> **Specific comments: Reviewer j9jR**
>
> We appreciate the time invested in carefully reading our paper and the very helpful and detailed
> comments. In particular, we thank you for pointing out that our work “is a very interesting manuscript and an enjoyable read”.
>
> 1.1) Non-negative definiteness of a matrix: We agree that the non-negative definiteness of a matrix does not apply when the matrix is not symmetric. Thank you for pointing this out, we will fix it in the revised version.
>
> 1.2) Gibbs distribution: We define a Gibbs distribution over all in-forests rooted at the seeds. In this case, the role of energy is played by the cost of the in-forests, which is equal to the sum of the cost of the edges in the forest. So there is an energy function in our approach, the map from in-forests to their cost. This Gibbs distribution is over in-forests while the probability distribution defined by the random walker is over absorbing directed walks. So they treat different and at first glance unrelated objects. We are able to show that the absorbing probability of a seed defined by the random walker starting at the query node (which is equivalent to the stationary probability of the random walker starting at the query node once all the seeds are set to absorbing nodes, which we assume you refer to) is equal to the probability, with respect to the Gibbs distribution, that a query node is part of the same tree of the seed in an in-forest. Our proof connects these two probability distributions which apparently are unrelated.
>
> 2) “Sampling directed graphs with random walks” by Ribeiro et al. aims to solve a different problem than we address. It focuses on the estimation of characteristics of directed graphs (e.g. the out-degree distribution). To do so, it samples nodes of the graph by means of a directed random walker. In contrast to our method, they only use one instantiation of the random walker, i.e. one walk. We instead consider all the possible walks to determine the absorbing probabilities we are interested in (though we approach it from the in-forests perspective). The problems are distinct in nature, since we assume that we have access to the complete underlying graph, while that is not the case in Ribeiro et al.’s work. Althotugh, it may be possible to apply our results to their problem, that would need further rumination. Thank you for pointing us out such an interesting paper.
>
> 3) Eigenvalues and MTT: Indeed, there are strong relations between the MTT and the eingenvalues of the Laplacian matrix in the undirected case. We are not aware of such a relation for the directed case. In the undirected case, all the determinants of the square submatrices of the Laplacian obtained by removing a fixed row and column $v$ are equal. These determinants coincide with the number of trees, independently of the choice of $v$. This fact facilitates the relation of the eigenvalues with the MTT. For the directed case, these determinants are not independent of the choice $v$ for the removed row and column which obstructs this relation. We would be interested if such a connection existed and plan to explore this in the future.
>
> 4) Out-forest extension: Indeed, all the results can be extended to out-forests. If one reverses the direction of all the edges of the graph, an in-forest transforms into an out-forest and vice versa. Therefore, applying our method for in-forests to the reversed graph, one can compute the probabilities for the out-forests. One could even combine the two points of view to improve the classification of the nodes.

---

### Author Response · Authors · 2021-08-10
**Experimental completeness (Reviewer ctCN, L2ar, CQ5E)**

We are thankful to all reviewers for the time and effort invested in carefully reading our paper and the very helpful and detailed comments. We will address their concerns in the revised version.

The numbered citations will refer to the references of the paper.

We agree that our experiments are not exhaustive enough to prove the practical superiority of the DProbWS method, but this was not our focus. Our aim was to emphasize that the extension to the directed setting of the ProbWS/Random Walker (RW) is possible. The RW algorithm is a very popular algorithm in the undirected case [3, 15, 43, 46]. The ProbWS paper [12] proposed a novel point of view of the RW by means of a Gibbs Probability distribution. In the presented work, we are able to bridge the Watershed and Random Walker realms for directed graphs, and show the community that such extension exists. Our work is valuable, as stated in the reviewer guideline, because it is a novel extension of well-known techniques. Proving equivalence is a crucial to keep the vastly expanding realm of machine learning structured, concise and manageable.

Nonetheless, we are happy to include more experiments for the revised version. In concrete, besides the ARW method [9], we run experiments comparing with the methods presented in [11,45], referred here as GTG and LLUD respectively. Moreover, we also added two citation networks: Cora [c] and CiteseerX [d].

Due to the high sparsity of the citation networks, many of the nodes are zero-knowledge nodes and therefore its label can not be inferred (see Remark 3.1). To remedy this, we make use of the so-called teleporting random walk (TRW), in the same way it is applied in [45], as a replacement of the natural one. In the TRW, a random walker jumps uniformly at random to any node with probability $\eta$, and with probability (1-$\eta$) takes an step of the natural random walker. This allows that any node can reach any other node. From the point of view of the in-spanning trees, this is equivalent to add to every node an out-going edge to all other nodes. The methods ARWtrw and DprobWStrw will refer to the versions that make use of the TRW.

In the tables below, we show the average accuracy (number of correctly labeled nodes divided by total number of nodes) over 5 runs and its standard deviation for different ratio label values $r$ between 0.1 and 0.9. The DprobWS and DprobWStrw method obtain comparative results for all datasets with respect to the other methods except for the EmailEU dataset, where they are worse. In the Cora dataset, DProbWStrw achieves the highest accuracy among all methods.

**Digits**

|r|0.1|0.2|0.3|0.4|0.5|0.6|0.7|0.8|0.9|
|---|---|---|---|---|---|---|---|---|---|
|DProbWS|0.978$\pm$0.005|0.987$\pm$0.001|0.991$\pm$0.002|0.992$\pm$0.003|0.993$\pm$0.001|0.994$\pm$0.001|0.996$\pm$0.001|0.997$\pm$0.001|0.999$\pm$0.001|
|DProbWStrw|	0.978$\pm$0.005|0.987$\pm$0.001|0.991$\pm$0.002|0.992$\pm$0.003|0.993$\pm$0.001|0.994$\pm$0.001|0.996$\pm$0.001|0.997$\pm$0.001|0.999$\pm$0.001|
|ARW|0.975$\pm$0.004|0.984$\pm$0.003|0.987$\pm$0.002|0.992$\pm$0.003|0.992$\pm$0.001|0.995$\pm$0.001|0.995$\pm$0.001|0.997$\pm$0.001|0.999$\pm$0.000|
|ARWtrw|0.942$\pm$0.007|0.55$\pm$0.005|0.960$\pm$0.004|0.974$\pm$0.003|0.976$\pm$0.003|0.980$\pm$0.001|0.987$\pm$0.002|0.992$\pm$0.002|0.995$\pm$0.002|
|GTG|0.978$\pm$0.006|0.986$\pm$0.004|0.99$\pm$0.003|	0.993$\pm$0.000|0.995$\pm$0.001|0.995$\pm$0.001|0.996$\pm$0.001|0.997$\pm$0.001|0.999$\pm$0.000|
|LLUD|0.915$\pm$0.014|0.968$\pm$0.003|0.983$\pm$0.002|0.988$\pm$0.001|0.988$\pm$0.001|0.989$\pm$0.002|0.99$\pm$0.001|0.991$\pm$0.001|0.991$\pm$0.000|

**20NewsGroups**

|r|0.1|0.2|0.3|0.4|0.5|0.6|0.7|0.8|0.9|
|---|---|---|---|---|---|---|---|---|---|
|DProbWS|0.653$\pm$0.016|0.75$\pm$0.004|0.806$\pm$0.007|0.849$\pm$0.003|0.883$\pm$0.003|0.912$\pm$0.003|0.938$\pm$0.002|0.96$\pm$0.002|0.981$\pm$0.001|
|DProbWStrw|0.663$\pm$0.024|0.75$\pm$0.005|0.807$\pm$0.007|0.85$\pm$0.003|0.883$\pm$0.003|0.912$\pm$0.003|0.938$\pm$0.002|0.959$\pm$0.002|0.981$\pm$0.001|
|ARW|0.654$\pm$0.018|0.743$\pm$	0.005|0.81$\pm$0.007|0.849$\pm$0.005|0.885$\pm$0.001	|0.913$\pm$0.004|0.938$\pm$0.002|0.96$\pm$0.002|0.981$\pm$0.001|
|ARWtrw|0.624$\pm$0.018|0.720$\pm$0.005|0.788$\pm$0.006|0.832$\pm$0.005|0.869$\pm$0.001|0.901$\pm$0.004|0.929$\pm$0.002|0.953$\pm$0.002|0.977$\pm$0.001|
|GTG|0.658$\pm$0.017|0.75$\pm$0.006|0.813$\pm$0.008|	0.854$\pm$0.004|0.885$\pm$0.002|0.914$\pm$0.003|0.940$\pm$0.002|0.960$\pm$0.001|0.981$\pm$0.001|
|LLUD|0.597$\pm$0.032|0.703$\pm$0.010|0.783$\pm$0.011|0.825$\pm$0.007|0.859$\pm$0.004|0.893$\pm$0.005|0.921$\pm$0.003|0.946$\pm$0.002|0.970$\pm$0.002|

**EmailEU**

|r|0.1|0.2|0.3|0.4|0.5|0.6|0.7|0.8|0.9|
|---|---|---|---|---|---|---|---|---|---|
|DProbWS|0.49$\pm$0.013|0.608$\pm$0.03|0.703$\pm$0.006|0.753$\pm$0.009|0.802$\pm$0.006|0.848$\pm$0.009|0.896$\pm$0.006|0.932$\pm$0.006|0.973$\pm$0.002|
|DProbWStrw|	0.49$\pm$0.014|0.617$\pm$0.030|0.703$\pm$0.006|.763$\pm$0.009|0.809$\pm$0.006|0.853$\pm$0.011|0.901$\pm$0.005|0.937$\pm$0.005|0.976$\pm$0.003|
|ARW|0.605$\pm$0.019|0.667$\pm$0.029|0.757$\pm$0.013|0.79$\pm$0.010|0.836$\pm$0.009|0.873$\pm$0.006|0.912$\pm$0.005|0.946$\pm$0.005|0.981$\pm$0.003|
|ARWtrw|0.574$\pm$0.012|0.639$\pm$0.014|0.716$\pm$0.005|0.760$\pm$0.008|0.806$\pm$0.008|0.848$\pm$0.007|0.896$\pm$0.005|0.933$\pm$0.005|0.974$\pm$0.002|
|GTG|0.611$\pm$0.016|0.671$\pm$0.007|0.722$\pm$0.007|0.768$\pm$0.005|0.807$\pm$0.007	|0.847$\pm$0.009|0.900$\pm$0.003|0.934$\pm$0.005|0.972$\pm$0.002|
|LLUD|0.575$\pm$0.012|0.639$\pm$0.016|0.719$\pm$0.005|0.764$\pm$0.009|0.809$\pm$0.008|0.851$\pm$0.009|0.900$\pm$0.005|0.9370.005|0.975$\pm$0.003|

**Cora**

|r|0.1|0.2|0.3|0.4|0.5|0.6|0.7|0.8|0.9|
|---|---|---|---|---|---|---|---|---|---|
|DProbWS|0.277$\pm$0.009|0.412$\pm$0.004|0.521$\pm$	0.007|0.616$\pm$0.009|0.702$\pm$0.008|0.770$\pm$0.003|0.836$\pm$0.002|0.892$\pm$0.004|0.949$\pm$0.003|
|DProbWStrw|0.456$\pm$0.009|0.574$\pm$0.003|0.655$\pm$0.007|0.7230.008|0.782$\pm$0.006|0.832$\pm$0.001|0.882$\pm$0.002|0.920$\pm$0.002|0.963$\pm$0.003|
|ARW|	0.185$\pm$0.010|0.317$\pm$0.015|	0.420$\pm$0.013|0.522$\pm$0.010|0.606$\pm$0.009|0.686$\pm$0.003|0.773$\pm$0.004|0.850$\pm$0.002|0.925$\pm$0.002|
|ARWtrw|0.376$\pm$0.007|0.445$\pm$0.003|0.518$\pm$0.003|0.589$\pm$0.003|0.657$\pm$0.004|0.729$\pm$0.003|0.795$\pm$0.004|0.866$\pm$0.003|0.933$\pm$0.003|
|GTG|0.209$\pm$0.001|0.301$\pm$0.002|0.396$\pm$0.003|0.485$\pm$0.002|0.572$\pm$0.003|0.662$\pm$0.005|0.749$\pm$0.004|0.834$\pm$0.004|0.920$\pm$0.002|
|LLUD|0.209$\pm$0.001|0.301$\pm$0.002|0.396$\pm$0.003|0.485$\pm$0.002|0.572$\pm$0.003|0.662$\pm$0.005|0.749$\pm$0.004|0.834$\pm$0.004|0.920$\pm$0.002|

**CiteseerX**

|r|0.1|0.2|0.3|0.4|0.5|0.6|0.7|0.8|0.9|
|---|---|---|---|---|---|---|---|---|---|
|DProbWS|0.260$\pm$0.010|0.404$\pm$0.011|0.497$\pm$0.011|0.601$\pm$0.008|0.682$\pm$0.003|0.758$\pm$0.005|0.830$\pm$0.004|0.889$\pm$0.003|0.946$\pm$0.002|
|DProbWStrw|0.376$\pm$0.020|0.491$\pm$0.010|0.572$\pm$0.011|0.662$\pm$0.008|0.730$\pm$0.003|0.794$\pm$0.005|0.854$\pm$0.004|0.907\$pm$0.004|0.954$\pm$0.002|
|ARW|	0.241$\pm$0.008|0.389$\pm$0.007|	0.509$\pm$0.005|0.596$\pm$0.006|0.681$\pm$0.004|0.755$\pm$0.001|0.822$\pm$0.003|0.887$\pm$0.002|0.947$\pm$0.002|
|ARWtrw|0.353$\pm$0.002|0.438$\pm$0.012|0.522$\pm$0.007|0.600$\pm$0.005|0.670$\pm$0.003|0.741$\pm$0.002|	0.081$\pm$0.004|0.874$\pm$0.001|	0.938$\pm$0.001|
|GTG|0.183$\pm$0.001|0.292$\pm$0.005|0.406$\pm$0.008|0.502$\pm$0.008|0.602$\pm$0.006|0.688$\pm$0.007|0.778$\pm$0.004|0.859$\pm$0.002|0.930$\pm$0.002|
|LLUD|0.404$\pm$0.006|0.510$\pm$0.016|0.594$\pm$0.010|0.674$\pm$0.009|0.740\$pm$0.005|0.800$\pm$0.004|0.858$\pm$0.003|0.908$\pm$0.002|0.954$\pm$0.002|

Though our experiments have been focused on semi-supervised learning, we believe that other applications can benefit from our results. Thanks to our theoretical work, the RW/ProbWS can be applied to a graph where the underlying adjacency matrix is not symmetric. This may be exploited in some applications where the weights are not symmetric, or in applications where the transitions are biased and occur more often in one direction than in the other. For instance, in the study of biological processes such as cellular differentiation and lineage choice. Given the single-cell RNA sequencing (scRNA-seq) of some developing cell population, the aim is to reconstruct the sequence of transcriptional modifications leading to potential cell fates. If we had access to the time point of measuring the genes for each cell, then one could construct a directed graph, which represents "likelihood of one cell state to evolve into another one". This might depend on the proximity in gene space, but also on the time direction, so that the graph is directed. Edges in opposite time direction are plausible, because cell development is not linear and we also measure the genes of parts of a population at different time steps (cells die once genes are measured), so instead of measuring one cell many times, we actually measure many cells only once, but at different time steps. For a pair of cells the arrow along the time direction will always have higher weight than the arrow against the time direction. The resulting graph would still have an overall "time-direction" and a random walker would eventually reach later time points, but not all edges point in the time direction and there might be cycles. Then, given some query cell, e.g. at a very early time step, one can use the DProbWS to compute how likely is it to end up in one of the final cell fates. Note that a vaguely similar approach was taken in [e].

[c] “Classification in networked data: A toolkit and a univariate case study” Macskassy et al., JMLR, 2007

[d] “The Network Data Repository with Interactive Graph Analytics and Visualization”, Rossi et al., AAAI, 2015

[e] “Generalizing RNA velocity to transient cell states through dynamical modeling” Bergen et al., Nature Biotech, 2020

---

> ### Author Response · Authors · 2021-08-10
> **Symmetrization of the kNN graph (Reviewer 1WWq, ctCN)**
>
> We believe that it may be very valuable information to know whether node $i$ is a k-nearest neighbour of $j$ but not vice versa. If that’s the case, it might mean, that $i$ lies in a sparser region of the graph.
>
> As suggested, we have run the DProbWS experiments after symmetrizing the kNN graph. Let $A$ be the non-symmetric adjacency matrix of the kNN graph. We have applied two kinds of symmetrization:
> - shared kNN graph (shKNN): in this case, the adjacency matrix of the symmetrized graph is defined as $\max(A,A^T)$, where $\max$ is the pointwise maximum operator.
> - mutual kNN graph (muKNN) [a,b]: in this case, the adjacency matrix of the symmetrized graph is defined as $\min(A,A^T)$, where $\min$ is the pointwise minimum operator.
>
> In the tables below, we show the DProbWS average accuracies (number of correctly labeled nodes divided by total number of nodes) over 5 runs and its standard deviation for different ratio label values between 0.1 and 0.9 and the different symmetrizations. As in the original paper, the zero-knowledge nodes (nodes which label can not be inferred, see Remark 3.1) are counted as incorrectly labeled. We observe that the DProbWS applied on the directed kNN graph (DirKNN) obtains better accuracy than the muKNN graph for both datasets (Digits and 20NewsGroups) but worse than the shKNN graph The increase of accuracy in the case of the shKNN graph is in part due to the reduction on the number of zero-knowledge nodes. Since the shKNN graph contains more “edges”, the connectivity of the graph is higher. When the graph is directed there exist more nodes that are not connected to the seeds via a directed path, and therefore the label of more nodes can not be inferred decreasing the accuracy.
>
> We have carried an extra experiment, where the adjacency matrix is not symmetric but still contains the same number of edges as the shKNN graph. To do so, we considered two graphs with the following adjacency matrices $\max(2A,A^T)$ and $\max(A,2A^T)$ which we refer as DirshKNN and DirshKNN$^T$ respectively. Since both directions of the edges are present, the reachability of the seeds by the nodes in the shKNN,  DirshKNN and DirshKNN$^T$ graphs are the same, and therefore both have the same zero-knowledge nodes. The difference between DirshKNN and DirshKNN$^T$ lies in that the first gives more weight to the directions present in the original kNN graph, while the second gives more weight to the ones not present in the DirKNN graph. We can observe that shDirKNN is marginally worse than shKNN, while shDirKNN$^T$ is slightly better than shKNN.
>
> Thus, we agree that the shKNN symmetrization on these concrete graphs would be meaningful, but still the non-symmetric graph shDirKNN$^T$ achieved better accuracies. Nonetheless, our aim was to illustrate the performance of the DProbWS in comparison to other methods designed for directed graphs, and not to compare the directed with the undirected setting.
>
>
> **Digits**
>
> |r|0.1|0.2|0.3|0.4|0.5|0.6|0.7|0.8|0.9|
> |---|---|---|---|---|---|---|---|---|---|
> |DirKNN|0.978$\pm$0.005|0.987$\pm$0.001|0.991$\pm$0.002|0.992$\pm$0.003|0.993$\pm$0.001|0.994$\pm$0.001|0.996$\pm$0.001|0.997$\pm$0.001|0.999$\pm$0.001|
> |muKNN|0.895$\pm$0.006|0.905$\pm$0.008|0.932$\pm$0.005|0.947$\pm$0.004|0.960$\pm$0.004|0.967$\pm$0.002|0.978$\pm$0.003|0.985$\pm$0.002|0.992$\pm$0.002|
> |shKNN|0.983$\pm$0.003|0.985$\pm$0.003|	0.991$\pm$0.002|0.992$\pm$0.003|0.993$\pm$0.000|0.994$\pm$0.001|0.996$\pm$0.001|0.998$\pm$0.000|0.999$\pm$0.001|
> |DirshKNN|0.980$\pm$0.003|0.985$\pm$0.003|0.991$\pm$0.002|0.992$\pm$0.003|0.993$\pm$0.002|0.994$\pm$0.000|0.996$\pm$0.001|0.998$\pm$0.001|0.999$\pm$0.000|
> DirshKNN$^T$|0.986$\pm$0.004|0.986$\pm$0.004|0.992$\pm$0.002|0.993$\pm$0.004|0.994$\pm$0.000|0.995$\pm$0.000|0.996$\pm$0.000|0.998$\pm$0.000|0.999$\pm$0.000|
>
>
> **20NewsGroups**
>
> |r|0.1|0.2|0.3|0.4|0.5|0.6|0.7|0.8|0.9|
> |---|---|---|---|---|---|---|---|---|---|
> |DirKNN|0.653$\pm$0.016|0.75$\pm$0.004|0.806$\pm$0.007|0.849$\pm$0.003|0.883$\pm$0.003|0.912$\pm$0.003|0.938$\pm$0.002|0.96$\pm$0.002|0.981$\pm$0.001|
> |muKNN|0.565$\pm$0.006|0.678$\pm$0.006|0.751$\pm$0.002|0.808$\pm$0.002|0.851$\pm$0.003|0.888$\pm$0.003|0.923$\pm$0.002|0.95$\pm$0.002|0.976$\pm$0.001|
> |shKNN|0.728$\pm$0.012|0.786$\pm$0.004|0.834$\pm$0.007|0.866$\pm$0.003|0.895$\pm$0.001|0.92$\pm$0.003|0.944$\pm$0.001|0.962$\pm$0.001|0.982$\pm$0.000|
> |DirshKNN|0.726$\pm$0.013|0.783$\pm$0.004|0.832$\pm$0.006|0.864$\pm$0.004|0.893$\pm$0.001|0.920$\pm$0.003|0.943$\pm$0.002|0.961$\pm$0.002|0.982$\pm$0.000|
> |DirshKNN$^T$|0.731$\pm$0.007|0.795$\pm$0.003|0.840$\pm$0.003|0.873$\pm$0.002|0.901$\pm$0.001|0.925$\pm$0.002|0.948$\pm$0.001|0.965$\pm$0.001|0.983$\pm$0.001|
>
>
>
> [a] “Using the Mutual k-Nearest Neighbor Graphs for Semi-Supervised Classification of Natural Language Data”, Ozaki et al., CoNLL, 2011.
>
> [b]“Connectivity of the mutual k-nearest-neighbor graph in clustering and outlier detection” Brito et al., Statistics & Probability Letters, 1997.

---

### Decision · Program_Chairs · 2021-09-27

**Decision:**

Accept (Poster)

**Comment:**

In this paper the authors proposed a semi-supervised learning algorithm for segmenting directed graphs, extending previously proposed methods designed  for undirected graphs The proposed method builds upon the watershed algorithm, an approached, based on building "separating forests"  and defining a Gibbs distribution, as in statistical physics, on this set of forests. The main contribution, beyond generalisation of this method to directed graphs, is given in the form  theoretical results related.

After the first round of rebuttal, the paper was found interesting and looked upon favourably on it theoretical part. The new theorems (for instance an equivalence with the random walker method) were all found strong and interesting by the reviewers. The numerical part, however, was found was rather weak and incomplete and the paper initially received mixed grading.

The authors did present a very good case in their rebuttal, giving new additional data and quantifying the comparative performance in their response to all the criticisms of the reviewers. After the rebuttal, it seemed that all reviewers (but one who did not reply, but whose criticism seemed to have been addressed) agreed that the paper was stronger and definitely of value to the Neurips community, actually increasing their score and assessment of the paper.

Given the authors successfully answered the criticism of the referee, I therefore recommend acceptance to Neurips.